# Use of a Molecular Switch Probe to Activate or Inhibit GIRK1 Heteromers In Silico Reveals a Novel Gating Mechanism

**DOI:** 10.3390/ijms231810820

**Published:** 2022-09-16

**Authors:** Dimitrios Gazgalis, Lucas Cantwell, Yu Xu, Ganesh A. Thakur, Meng Cui, Frank Guarnieri, Diomedes E. Logothetis

**Affiliations:** 1Department of Pharmaceutical Sciences, School of Pharmacy and Pharmaceutical Sciences, Bouvé College of Health Sciences, Boston, MA 02115, USA; 2Center for Drug Discovery, Northeastern University, Boston, MA 02115, USA; 3Department of Chemistry and Chemical Biology, College of Science, Boston, MA 02115, USA; 4Roux Institute, Northeastern University, Portland, ME 04101, USA

**Keywords:** GIRK1 heterotetramer, ML297, MD simulations, PIP_2_, TM1 hydrophobic wire

## Abstract

G protein-gated inwardly rectifying K^+^ (GIRK) channels form highly active heterotetramers in the body, such as in neurons (GIRK1/GIRK2 or GIRK1/2) and heart (GIRK1/GIRK4 or GIRK1/4). Based on three-dimensional atomic resolution structures for GIRK2 homotetramers, we built heterotetrameric GIRK1/2 and GIRK1/4 models in a lipid bilayer environment. By employing a urea-based activator ML297 and its molecular switch, the inhibitor GAT1587, we captured channel gating transitions and K^+^ ion permeation in sub-microsecond molecular dynamics (MD) simulations. This allowed us to monitor the dynamics of the two channel gates (one transmembrane and one cytosolic) as well as their control by the required phosphatidylinositol bis 4-5-phosphate (PIP_2_). By comparing differences in the two trajectories, we identify three hydrophobic residues in the transmembrane domain 1 (TM1) of GIRK1, namely, F87, Y91, and W95, which form a hydrophobic wire induced by ML297 and de-induced by GAT1587 to orchestrate channel gating. This includes bending of the TM2 and alignment of a dipole of two acidic GIRK1 residues (E141 and D173) in the permeation pathway to facilitate K^+^ ion conduction. Moreover, the TM movements drive the movement of the Slide Helix relative to TM1 to adjust interactions of the CD-loop that controls the gating of the cytosolic gate. The simulations reveal that a key basic residue that coordinates PIP_2_ to stabilize the pre-open and open states of the transmembrane gate flips in the inhibited state to form a direct salt-bridge interaction with the cytosolic gate and destabilize its open state.

## 1. Introduction

Gating mechanisms of ion channel proteins have remained elusive, despite recent great advances in structural biology. The challenge is that gating is a highly dynamic process and not enough frames in the “gating movie” have been captured yet. Computational approaches capable of providing atomistic details of the dynamics of state transitions from a closed to an open conformation have been challenging to attain, requiring computationally expensive simulations. We have pursued this problem using the G protein-gated K^+^ channels for which much structural information has been achieved and which have been studied extensively both experimentally and computationally. There are four members in the Kir3 (or GIRK for G protein-gated inwardly rectifying K^+^) channel subfamily, GIRK1-4. The GIRK family members are highly conserved (50–70% identity). All subunits are expressed in the brain and neuronal tissues with GIRK1 and GIRK2 being the most prevalent. GIRK1 and GIRK3 are not functional as homotetramers but upon heteromerization, with GIRK2 or GIRK4 or with each other, they produce conductive channels with unique properties [1]. GIRK2 and GIRK4 can also function as homotetramers. The GIRK1 and GIRK4 channels are also expressed in non-neuronal tissues, the best characterized of which are the atrial GIRK1/4 heterotetramers. The physiological regulators of these channels are the Gβγ subunits of heterotrimeric GTP-binding (Gαβγ) proteins [2,3] that interact directly with the channels to activate them at nanomolar concentrations. In addition, concentrations of tens of millimolar intracellular Na^+^ ions entering through Na^+^-conductive ion channels can potentiate channel activity in a manner independent of Gβγ activation [4,5]. Yet, Na^+^ and Gβγ synergize in activating the channel [5]. Kir3 channels were shown to have an absolute dependence on phosphatidylinositol bis-phosphate (PIP_2_) for activation [4,6]. Since PIP_2_ and either Na^+^ or Gβγ are needed for gating and all three are needed for optimal gating, they are often referred to as gating co-factors for GIRK activity. Atomic resolution structures of GIRK domains and functional channels, especially GIRK2, alone or with regulators, have been generated by the MacKinnon lab since 2002 [7,8,9,10,11]. These structures have not only provided a wealth of information of static snapshots of GIRK channels in isolation or together with their regulators but have also enabled dynamic simulations to provide insights into their functional properties.

The tetrameric GIRK2 channels are composed of three protein domains, the transmembrane domain (TM1 or outer and TM2 or inner helix), and the cytosolic domains, the N-terminal domain (NTD), and the C-terminal domain (CTD) (Appendix A). The TM2 helix forms the central pore of the ion channel extended by the cytosolic domains (permeation pathway highlighted in gray) (Appendix A). In this helix, a set of four highly conserved phenylalanine residues (GIRK2-F192) contributed by each subunit can constrict the inner pore and prevent ion conductance. This constriction (or gate) is referred to as the helix bundle crossing (HBC) (Appendix A). The second gate is formed by the cytosolic extension by two Met residues (GIRK2-M313/M319) resembling a Girdle and thus named the G-loop gate (Appendix A). A highly conserved i−12 glycine residue (G180 relative to the F192 HBC gate) can act as a hinge to allow for pore dilation and channel conduction [12]. Substitution of the i−11 residue (S181) with a proline, kinks the TM2 helix outwardly and induces pore dilation in GIRK channels [12,13]. The endogenous regulators of GIRK channels (Na^+^, Gβγ, and PIP_2_) can couple onto distinct gates to induce channel opening [14]. The Gβγ dimer predominantly works to induce the opening of the HBC. Gβγ binding induces a rocking motion of the cytoplasmic domain bending the two transmembrane helixes and inducing the pore dilation at the HBC. The i−11 serine to proline mutation (S181P in GIRK2 or S176P in GIRK4) makes the channel unresponsive to Gβγ dimer stimulation suggesting that the induced bending motions of transmembrane helix 2 lead to HBC opening. This prior work is foundational to informing the mechanism involved with the opening of this channel.

A recent series of pharmacological tools have been produced that can gate GIRK channels to the open or closed conformations independently of other physiological regulators (e.g., Gβγ or Na^+^), but their dependence on PIP_2_ remains. A class of small molecule GIRK channel activators, with ML297 as the prototype, serve as potent activators selective for GIRK1-containing channels [15,16,17,18,19,20]. ML297 activates GIRK1-containing channels requiring only two amino acids specific to GIRK1, F137, and D173 [19]. Although ML297 was shown to be more biased towards GIRK1/2 activation than GIRK1/4 (~10-fold), it can be considered relatively nonselective (or nonspecific) when compared to the selective (or specific) GAT1508 (100-fold preference for GIRK1/2 over GIRK1/4) [19,20]. Thus, the activation of GIRK1/4 expressed abundantly in supraventricular cardiac myocytes has limited the utility of ML297 as a potential drug targeting GIRK1 heteromers expressed in the brain. We find this class of small molecules fascinating probes to study molecular control of channel activity and we have been successful in producing a completely specific GIRK1/2 activator, GAT1508, with no cardiac side effects [20]. Figure 1A shows the basic structure of ML297, consisting of a required urea core (Site 2) flanked by a difluoro-phenyl moiety (Site 1) on one end, where alternate substituents can be tolerated, and a pyrazole moiety (Site 5) on the other end with benzyl/non-aryl moieties on the 5-position of the pyrazole ring (Site 4) and cycloalkyl groups on the 3-position of the pyrazole ring (Site 3). Wen and colleagues [17] discovered that various functionalized 3-cyclopropyl moieties at Site 3 switched the mode of pharmacology of ML297 from activator to inhibitor. Here, we became interested in exploring the molecular mechanism of the Site-3-dependent molecular switch in computational models of GIRK1-containing channels (Figure 1B,C), as such a minimal substitution could be used as a surgical mechanistic probe to study the channel transition from an inhibited to an activated conformation. We synthesized GAT1587, a methyl cyclopropyl Site-3 variant that behaved as a non-specific inhibitor of GIRK1/2 and GIRK1/4 channel activity switched from the non-specific activation obtained by ML297 (Appendix A) [17].

## 2. Results

### 2.1. Sub-Microsecond Long Stochastic Dynamics Simulations Capture Ligand Effects, Validating the Utility of the Computational Model

ML297 and GAT1587 are from a concerted series of compounds that utilize the same core scaffold. This scaffold is composed of three distinct regions: a 3,4-difluorophenyl head group (Site 1), a urea linker (Site 2), and an N-phenyl pyrazole core (Sites 3–5). The major distinction between these compounds comes in Site 3 from a 3′ substitution on the pyrazole ring that serves as a molecular switch to control the pharmacological activity of these compounds (Figure 1A). The addition of a methyl cyclopropyl group in ML297 can invert the pharmacological effect of the compound from activation to inhibition of GIRK1/X activity (Appendix A and ref. [17]). Thus, while ML297 is an activator of GIRK1/2 and GIRK1/4 heterotetrameric ion channels, GAT1587 inhibits the same set of channels (Appendix A). The mechanism by which the presence of the methyl cyclopropyl moiety switches the effect of the compound from activation to inhibition in these channels remains unidentified.

To pursue the molecular switch mechanism of action, we constructed homology models of GIRK1/2 and GIRK1/4 channels based on the crystal structure of the preopen structure of GIRK2 (4KFM, see Section 4). Two ML297 or GAT1587 compounds were docked onto the two GIRK1-containing heterotetramers using an induced-fit docking protocol in the absence of the required co-factor PIP_2_.

The two unique GIRK1 residues, Phe137 and Asp173, were shown to be necessary and sufficient to confer ML297 sensitivity to GIRK1-containing heterotetramers [19]. In fact, the corresponding GIRK2 double mutant Phe148 and Asp184 (GIRK2-FD) behaved as a GIRK1-like subunit showing great sensitivity to ML297 in GIRK2/GIRK2-FD heterotetramers compared to the insensitive GIRK2 homotetramers [19,20].

The binding site for the urea scaffold-containing compounds that is partially composed of GIRK1 residues Phe97 and Phe175 was initially validated for a related GIRK1/2 (and GIRK2/GIRK2-FD) selective activator GAT1508 [20]. While GAT1508 is more selective than the parent ML297 in activating GIRK1/2 over GIRK1/4, these compounds only differ in Site 1, where the difluorophenyl ring is replaced with bromothiophene. The binding sites of these two compounds are identical on GIRK1/2. In GIRK1/4 as well as the corresponding GIRK4/GIRK4-FD, GAT1508 occupies the same binding site although in a flipped conformational pose [20].

This provided a reasonable set of starting coordinates for model generation. These resulting systems were solvated in a mixed membrane bilayer composed of POPC, POPE, POPS, and cholesterol. Finally, 150 mM KCl was added to the systems representing a “native” potassium ion concentration. Stochastic dynamics simulations under an applied electric field of −0.06 kcal/(mol*A*e) were used to simulate the native electrochemical gradient for potassium ions. Under these conditions, if a compound is to shift from the initial preopen state of the channel to an open or closed form, the resulting ion conduction should be representative of the functional state of the channel. However, this is only one of several collective variables that can be used to analyze these systems.

We defined three key collective variables that are critical to the gating and permeation through these K^+^ channels: (a) the normalized salt bridge formation between a basic lysine residue of the channel-PIP_2_ binding site and PIP_2_ that is considered a key step controlling gating; (b) the minimum distances of the helix bundle crossing and G-loop gates that allow permeation; and c) the conduction of potassium ions through the channel. We ran multiple replicas of simulations and generally presented the results from a representative run (e.g., Figure 1 and Appendix A) but we also performed statistical analysis to assess variability among replica runs (i.e., Appendix A). Urea-based compounds, such as ML297 or GAT1508, have been shown to allosterically modulate GIRK1-containing heterotetrameric ion channels [19,20]. In studies utilizing light-activated or voltage-activated phosphatases to deplete PIP_2_ from the membrane, dosing with ML297 or GAT1508 reveals the dependence of urea-based drugs on channel–PIP_2_ interactions to exert their effects [19,20]. The channel-PIP_2_ affinity can be represented through the average salt bridge formation between the P4 and P5 phosphates of PIP_2_ with critical basic lysines (GIRK1: K183, K188, and K189; GIRK2: K194, K199, and K200; GIRK4: K189, K194, and K195). While this serves as a good indication of compound activity, the simulations could result in a configuration where the normalized salt-bridge formation is increased but in an unproductive manner that does not lead to channel opening. To address this, we also monitored the size of the helix bundle crossing and G-loop gates. This allows for direct measurement of the conformational state. However, it has been reported in the literature that while gate opening is necessary for ion conduction, it alone is not sufficient [21]. As a result, we also chose to monitor the number of ions conducted through the channel throughout the last 150 ns of each simulation. Ion conduction was defined as a single ion transversing both sets of gates within the channel. In the second half of the simulations, activated channels were seen allowing ions transverse the full length of the channel. Using this approach, we cross-validated potential conformational changes during replicate simulations and reported them accordingly as part of this work.

Running simulations of these channels for a total of 300 ns provides ample time not only for the channels to equilibrate but for them to transition into a functional state. It should be noted that ML297 and GAT1587 undertake distinct equilibrium binding poses. ML297 tends to favor intercalating between TM1 and TM2 of the GIRK1 subunit (Figure 1B-left). In comparison, GAT1587 binds along with the transmembrane interface rather than intercalating between the helices (Figure 1B-right). This difference in the binding pose is due to the added hydrophobic bulk from the molecular switch, i.e., the addition of the methyl cyclopropyl group on Site 3.

Examination of the three collective variables in the simulations of the GIRK1/2 heterotetramer reproduces the pharmacological effects of ML297 and GAT1587. ML297-induced ion conduction over the last 150 ns of the simulation when compared to the Apo PIP_2_-only complexed channel (Figure 1D). Similarly, GAT1587 completely prevented any ion conduction during the simulation (Figure 1D). Both intermolecular-gate distances behave consistently with what is seen with ion conduction. ML297 causes a significant dilation from the 5.7 Å minimal diameter that occludes permeation of partially solvated potassium ions through both gates and especially the G-loop gate (Figure 1E). As having both gates open is a prerequisite for ion conduction, the “gate distance” collective variable underlies the resultant ion conduction (Figure 1D). GAT1587 induces a constriction of the G-loop gate (Figure 1E). Finally, comparable pharmacological activities are demonstrated through the normalized salt-bridge formation with PIP_2_ (Figure 1F), consistent with the view that the enhanced salt-bridge formation with PIP_2_ opens both gates and allows for ion conduction, while the decreased salt-bridge formation with PIP_2_ collapses at least one of the channel gates (the G-loop gate in this case) to prevent ion conduction.

Due to the non-specific nature of ML297 and GAT1587 (i.e, they activate and inhibit both GIRK1/2 and GIRK1/4 heterotetramers), we see the same behavior when these compounds are complexed with GIRK1/4. ML297 and GAT1587 demonstrate similar preferences in equilibrium binding poses (Appendix A). In the case of ML297, this results in enhanced ion conduction, dilation of both the G-loop and helix bundle crossing gates, and an increase in normalized salt bridge formation between the channel and PIP_2_ (Appendix A). GAT1587 inhibits ion conduction throughout the simulation, produces constrictions in both the G-loop gate and the helix bundle crossing, and a decrease in the normalized salt bridge formation between the channel and PIP_2_ (Appendix A). These findings validate the use of these collective variables in defining the functional states of the channels in silico. While other interesting states of these channels associated with these compounds are likely to exist, we have selected to analyze the three states shown due to their pharmacological relevance.

### 2.2. Ligand Binding to GIRK1 Subunit of Heterotetramers with GIRK2 or GIRK4 Reveals Key Differences between Residue Conformations

Since these urea-based compounds bind GIRK1 specifically, we docked two compounds per heterotetramer GIRK1/2 or GIRK1/4 (docking poses of starting simulations are shown in Appendix A). Simulations were run with these pre-docked compounds to obtain the results discussed above in Figure 1 and Appendix A. Within 50 ns of the MD runs equilibration was reached as can readily be seen in the RMSD of the backbone C-α carbons in multiple replica runs of GIRK1/2 and GIRK1/4, each with either the activator (ML297) or the inhibitor (GAT1587) (Appendix A). Given the nonspecific nature of ML297 and GAT1587 and their dependence on GIRK1 for activity, we might have expected these compounds to bind primarily to the GIRK1 subunit and be insensitive to the partner subunit. In conducting a more detailed binding site analysis of ML297 on GIRK1/2 (Figure 2A,B), we see that its intercalation between the two transmembrane domains is driven by three different sets of interactions. (a) GIRK1-Q165 coordinates with the indispensable central urea linker in Site 2 by presenting the carbonyl of the side-chain amide group towards the urea linker, resulting in the creation of two hydrogen bonds. It should also be noted that these hydrogen bonds are shielded by the hydrophobic residues that line each transmembrane helix; (b) the 3,4 difluorophenyl head group in Site 1 can pi stack (or T stack) with a neighboring GIRK1-F164; (c) the interactions position the phenyl ring of the n-phenylpyrazole portion of the compound in such a way to Pi stack with GIRK1-W95 in Site 4, while GIRK1-M98 bridges Site 1 and Site 4 (see Figure 2B). These interactions are nearly identical to those obtained when ML297 binds onto the GIRK1/4 heterotetramer (Appendix A). ML297 maintains the same equilibrium binding pose on either ion channel heterotetramer, which is consistent with the non-specific nature of this compound.

GAT1587 on the GIRK1/2 does not make the same set of hydrogen bonds with GIRK1-Q165 as ML297 does due to its binding along the transmembrane helices rather than intercalating between them (Figure 2C,D compared to Figure 2A,B). This compound’s interactions are driven through a set of pi sulfur or pi–pi interactions between Site 1, the 3,4-difluorophenyl head group, and the GIRK1-M98 that no longer bridges with Site 4 and F164, respectively. The N-phenyl pyrazole groups (Sites 4 and 5) engage in a series of alkyl or pi-alkyl contacts with GIRK1-A94 or V172. Finally, the methyl cyclopropyl molecular switch stabilizes a flipped conformational pose through a series of hydrophobic interactions with the GIRK2 subunit residue I195 and W91 (Figure 2C,D). While most of these interactions are preserved when GAT1587 binds onto the GIRK1/4 heterotetramer (Appendix A), there are some differences between the binding modes. Notably, the methyl cyclopropyl ring still interacts with GIRK4-W86 (like with the GIRK2-W91) but the conformations between GIRK2-W91 and GIRK4-W86 do differ. Moreover, the other interactions of the cyclopropyl ring seen with the GIRK1/2 subunits (I195.2 and I171.1 and F175.1) are missing in the GIRK1/4 heterotetramer. Otherwise, the overall binding mode of GAT1587 is similar in the presence of GIRK2 or GIRK4.

### 2.3. Ligand Activation Induces a Hydrophobic Chain of Residues in TM1 That Relieves Its Restrain on TM2, Allowing It to Bend and Stabilize the HBC Gate in the Open State

With the functional effects of the compounds reproduced by the sub-microsecond computational models, we could begin to identify the underlying mechanism that produced these effects. Prior work has strongly implicated TM2 bending in the mechanism of channel activation [12,14]. Within the Kir3 family, there is a conserved glycine residue, GIRK4-G175 (where it was first studied, GIRK1-G169, GIRK2-G180), which serves as a critical hinge point within TM2. Work conducted on GIRK4 homotetramer revealed that substituting a proline at the i+1 position, GIRK4-S176P induced a bending moment in TM2. This increased the open-channel probability in the absence of other activators such as Gβγ [22]. Similarly, if the conserved glycine was mutated into a threonine, the beta branch could serve as a directing group resulting in a more extreme bending moment on TM2. This resulted in an even further increase in channel-open probability [12]. The bending moment around TM2 provides a starting point for further analysis. We hypothesized that ML297 and GAT1587 exerted their activation or inhibition, respectively, by bending TM2 at the conserved Gly residue but in opposite directions.

To monitor a possible bending moment on TM2, we defined two vectors to represent the different segments of the transmembrane helix. In this case, we could use the center of mass of the backbone atoms to define the center of the helix at the central bending pivot and the two termini, the outer and inner portions of the helix. In this case, the central bending point of the helix was formed by GIRK1 residues I167 to G169. We could use one turn above (G158 to F162) and one below (A174 to G178) relative to the center pivot point to define the axes of the outer and inner portions of the TM2 helix, respectively (Figure 3A). With this framework, we monitored the effects of both compounds on the bending of TM2.

It should be noted that while the conserved GIRK1 residue G169 is one of the major pivot points, it is not the only one. The GIRK1-I167 also serves as a second pivot point. This residue undergoes a conformational change due to being placed under tension rather than compression-like GIRK1-G169. The backbone dihedrals of the particular residue change over 20 degrees shifting the dihedrals from the favorable region into the allowed region in the Ramachandran plot (Figure 3C).

Finally, using the vectors previously described, we can quantify the bending moment that is placed on TM2 around the conserved GIRK1-G169 (Figure 3D1,D2). While the Apo system produced a bend between the two transmembrane helix segments of approximately 170 degrees, the activator ML297 exacerbates the native bending motion decreasing the angle to ~165 degrees. In comparison, GAT1587 almost linearizes TM2 and results in an angle of ~177 degrees. The activity of these compounds can be thought of either as the result of imparting a bending moment directly onto TM2, due to binding near the conserved GIRK1-G169 pivot point, or by controlling a restraint potential that is placed onto the core TM2 bundle by the outer TM1 bundle. We can quantify the inter transmembrane helix contacts through a coarse grain analysis for a restraining potential that TM1 exerts on TM2. We monitored the total contacts made between the TM1 and TM2 on a per-atom per-frame basis. These contacts were then summated over the frame to give the total contacts made between the two TM domains. ML297 causes a noticeable decrease in the total contacts that are made between the two transmembrane helixes (Figure 3E1,E2). GAT1587 on the other hand has the opposite effect, causing a dramatic increase in the total contacts made by the two transmembrane helices when compared to the apo state. This suggests that these compounds act by directly modulating the restraining potential that TM1 places on TM2. The modulation of the restraining potential also correlates with an outward movement of TM1 in the presence of the activator ML297 relative to GAT1587, a difference that is not seen in the overall change of TM2 (Figure 3F).

To analyze the underlying atomistic changes that drive these conformational changes, we performed a shortest pathway analysis to build a correlated network of residues from the ML297/GAT1587 binding sites to channel sites shown to be directly involved in gating, such as the CD loop (controls the G-loop gate) or to the PIP_2_ binding site (directly controlling the HBC gate). While these networks are similar, being primarily composed of residues that make up TM1, the SH, the B loop, and CD loop regions, GAT1587 showed a greater degree of pathways that bridge between the two transmembrane domain regions than ML297 (Appendix A). This agrees with the hypothesis that TM1 acts as a restraint on the movement of the TM2 bundle. This shortest pathway analysis revealed the greatest effects in two distinct regions.

The first region that was identified through the shortest pathway analysis is a highly correlated network of residues that line TM1. This series of residues is composed of two separate but important components. The first component is primarily the i−4 backbone hydrogen-bonding network that is present in alpha helices. This hydrogen-bonding network allows the helix to maintain its structure and bend as a unit. Secondly, we also identified the creation of an extended pi system that couples to the pore-lining helix and TM2 at the level of the HBC gate (GIRK1-F181) through interactions with GIRK1-M180. This extended pi system, which we will refer to as the “TM1 hydrophobic wire”, is centered around a highly conserved residue within the Kir3 family, GIRK1-Y91 (Tyr or Phe), which neighbors two other conserved residues within the Kir superfamily, GIRK1-F87, and W95. This sequence of residues, at the i−4, i, and i+4 positions, allows for the formation of a sequence of pi–pi interactions to form parallel to the axis of the helix. This TM1 hydrophobic wire through extended pi interactions forms the backbone of a system that is meant to induce charge delocalization on neighboring residues to allow for ion conduction. The GIRK1-specific F137 also plays a role in this extended pi system. This unique GIRK1 residue has long been recognized to boost the activity of other GIRK channels that have a Ser residue at the corresponding position (e.g., GIRK4 or GIRK2) [23,24] and has also been shown to be partially responsible for the GIRK1 specific effect of ML297 [19]. However, its precise role within channel activation is yet to be determined.

This extended pi system can be broken up into three different components that make up the upper portion of TM1, a portion of the pore-lining helix, and related residues on both the GIRK1 and GIRK2 TM2. The core of TM1 hydrophobic wire is built on a series of pi-stacking interactions that form between GIRK1-F87, GIRK1-Y91, and GIRK1-W95 in the ML297 activated channel. GIRK1-W95 also interacts with the GIRK1 specific F137 from the pore-lining helix through T stacked pi interactions. Finally, GIRK1-F87 is also flanked by GIRK1-M180, which is located next to F181, the GIRK1 HBC gate, which can interact with the extended pi system through either a pi–sulfur interaction due to the relatively electron-rich phenyl rings or through Van der Waals effects (Figure 4A).

ML297 and GAT1587 have contrasting effects on the formation of this extended pi system. The interaction between GIRK1 residues W95-Y91, Y91-F87, and F87-M180 are all dependent on the state of the channel and can be induced in the presence of ML297. Pi-stacking interactions are typically on the order of fewer than 4 angstroms for these interactions (Figure 4B). The GIRK1 specific F137 is not sensitive to the presence of either ML297 or GAT1587 as revealed by these sets of simulations. This residue will T stack with GIRK1-W95 irrespective of the activating or inhibiting compound. GIRK1-M180 on the other hand is actively recruited into this system in the presence of ML297.

The bending moment that is imparted on TM2 as discussed above (Figure 3) forces GIRK1-M180 closer to GIRK1-F87 and results in stabilizing the TM2 helix in a bent conformation.

Conversely, GAT1587 results in a significant disruption to the pi-stacking interactions of the TM1 hydrophobic wire with typical distributions on the order of 5 to 6 angstroms (Figure 4B). GIRK1-M180 is also significantly disrupted when GAT1587 binds and results in a distance of over 6 angstroms (Figure 4B). This interaction distance would render both pi-sulfur and Van der Waals interactions negligible.

While the GIRK1 M180 might be engaging with the pi system through quantum mechanical pi–sulfur interactions or conventional steric bulk, there is additional steric bulk that can line the pi-pi stack to help stabilize the conformation of each residue. Each member of the TM1 hydrophobic wire recruits an additional hydrophobic residue that comes from an adjoining GIRK2 subunit (Figure 4C). The upper cut off for the hydrophobic interactions is defined as 3.0 angstroms from the atom centers. While no general value for these interactions has been suggested, carbon, nitrogen, and oxygen have Van der Waals radii ranging from 1.5 to 1.7 angstroms [25]. If we consider that these distance measurements are taken from the atom centers, doubling the smallest Van der Waals radius gives a conservative estimate for the upper cut-off of ~3.0 angstroms.

GIRK2-L174, V178, and I182 provide additional steric bulk that can restrict the conformation of the TM1 hydrophobic wire in the open state. GIRK2-L174 can interact with GIRK1 W95 through Van der Waals. This interaction was slightly enhanced in the presence of ML297 (Figure 4D). In comparison, the Van der Waals interactions between GIRK2-V178 and GIRK1-Y91 were essentially insensitive to the presence of the compound.

However, GIRK2-I182 could make prominent Van der Waals interactions with GIRK1-F87. Thus, in the active state, GIRK1-F87 recruits GIRK2-I182 in addition to GIRK1M180 (discussed above) to stabilize the lower portion of the TM1 hydrophobic wire.

This recruitment could be facilitated in part due to the bending moments imparted onto TM2 by ML297. These interactions are relatively short-ranged even for Van der Waals interactions being on the order of 2 angstroms in the open state (Figure 4D). Conversely, this recruitment is not present in the GAT1587-mediated closed state.

### 2.4. The TM1 Hydrophobic Chain Regulates K^+^ Ions at the E141-D173 Acidic Residue Pair of the Permeation Pathway, Depending on the Ligand Bound to the Channel

While we have described the formation of the TM1 hydrophobic wire, we have yet to identify the role it plays in ion conduction. Further comparison of residues within the hydrophobic wire exhibiting differential interactions in the presence of ML297 versus GAT1587 link the hydrophobic TM1 wire to ion conduction. The formation of the TM1 hydrophobic wire positions the indole hydrogen of GIRK1-W95 such that it participates in charge dipole-type hydrogen bonds with GIRK1-E141 (Figure 4E1,E2). This interaction is not possible in the presence of GAT1587 (Figure 4F). The GIRK1 residue E141, which is conserved only within the Kir3 subfamily, is another pore helix (PH) (F130-A142) residue (besides the GIRK1 specific F137 residue) that influences channel activity. This residue has also long been shown experimentally to promote channel activity in GIRK2 (E152D [26]). Furthermore, pre-existing computational evidence also suggests a critical role for the GIRK2-E152 in ion conduction work that will be elaborated further in the context of our findings in the discussion [21]. On the other hand, GIRK1-Y91, the residue neighboring W95 in the TM1 hydrophobic wire, is fully engaging through hydrogen bonding another GIRK1-unique acidic residue among GIRKs, D173, an interaction that is considerably weakened in the presence of ML297. Two additional hydrogen-bonding interactions are relieved in the presence of ML297 compared to GAT1587. Not only, as we discussed above, does ML297 establish the GIRK1-F87 (third residue of the TM1 hydrophobic wire) to interact with the GIRK2-I182 in TM2 but GAT1587, which breaks this interaction, establishes a GIRK1-Y91 interaction with the adjacent GIRK2-S181 residue, next to the flexible hinge GIRK2-G180 residue. Moreover, the GIRK2-S181.2 residue hydrogen bonds with the GIRK1-D173, preventing it from coordinating K^+^ ions in the permeation pathway. This interaction is broken in the presence of ML297, allowing D173 to facilitate ion conduction. D173 was identified as the second critical GIRK1 residue (besides F137) for ML297-mediated channel activation [19]. Moreover, this acidic residue at this position has been shown to play a strong role in Kir channel rectification [27]. These connections of TM1 hydrophobic wire residues to the GIRK1-E141 and D173 residues implicated the marked dissociation of TM1 and TM2 to the activated form of the channel. Our data also suggest that the engagement of E141 by W95 and the liberation of D173 from Y91 (directly and indirectly via the GIRK2-S181) achieved by ML297 and reversed by GAT1587 are critical for ion conduction (Figure 4G). In the discussion section, we propose a model of how these two acidic GIRK1 residues could be coupled to serve a critical role in ion conduction.

### 2.5. Upon ML297 Binding, TM1 Movement Is Transduced to the Slide Helix (SH) and the CD Loop, Freeing GIRK1-K188 Away the GIRK2-E315 (G-Loop) and toward PIP_2_ Binding

Given the conformational changes of TM1 (W82-I105) as the channel undergoes activation, we looked for conformational changes induced to the slide helix (SH) (Y67-V76) region that is directly linked to TM1 through a highly conserved 5-aa loop (D77-R81) that contains the K/R/Q-W-R motif that coordinates the P1 phosphate of phosphoinositides. The SH region has been implicated in the control of the G-loop gate through the CD-loop, where Na^+^ ions act, as well as to be critically coordinating the mechanism of Gβγ activation of the HBC gate [14]. To identify changes to the SH region, we defined two vectors that are parallel with the axis of TM1 and SH regions. When measuring the angle between these two vectors, it becomes clear that ML297 and GAT1587 have contrasting effects on this region as well. ML297 induces an upward movement of the SH region which, as Figure 5A shows, results in the angle between the TM1 and the SH becoming more acute (45 to 35 degrees). The opposite effect (45 to 50 degrees) is seen with GAT1587, making the TM1-SH angle more obtuse. Superposition relative to the GIRK1 TM1 makes the relative motions of the SH region clear (Figure 5B).

The shortest pathway analysis identified the SH as a second region (besides the TM1 hydrophobic wire) where a highly correlated network of residues from the ML297/GAT1587 binding sites to SH channel sites was directly involved in the differential gating by the two drugs. The SH region is composed of a series of residues that interact with the CD loop to control the activation of the G-loop gate [14]. Of the GIRK2 CD loop, GIRK2-R230 is an important residue for activation of the G-loop gate by transducing interactions of the SH through the CD loop [14]. GIRK2-R230 forms salt-bridge interactions state-dependently with two negatively charged aspartates in the SH region, GIRK1-D70 and D77. In the ML297-mediated active state of the GIRK1/2 channel, GIRK2-R230 forms a salt bridge with GIRK1-D77 after dissociating with GIRK1-D70 (Figure 5C,D). For comparison, the GAT1587-mediated closed-channel state greatly enhances the GIRK2-R230 salt bridge with GIRK1-D70. In the inhibited state, this interaction is on the order 2 angstroms (Figure 5D). It should be noted that the dissociation of GIRK2-R230 from GIRK1-D70 is due to two important events. In one case, the separation of the transmembrane helices forces a change in the bending angle between TM1 and the SH. As the angle between TM1 and the SH becomes more acute, the SH and CD loop, and, thus, GIRK2-R230 and GIRK1-D70, begin to separate from each other. However, due to the relatively strong salt bridge that is present in the apo and inhibited cases, which is typically less than 4 angstroms, the GIRK1 residues T73 and T74 are required to sequester off the charged side of GIRK1-D70. These 2-residue quasi-covalent hydrogen bonds with GIRK1-D70 are on the order of 2 angstroms (Figure 5C,D). These tight sets of hydrogen bonds can facilitate the delocalization of the negative charge of GIRK1-D70. This allows for the dissociation of GIRK1-D70 with GIRK2-R230 and the formation of a novel salt bridge between GIRK2-R230 and GIRK1-D77. The switching motif of GIRK2-R230 has further effects that propagate from the CD loop to the G-loop gate and ultimately result in the opening of this gate and the channel overall. The opening of the G-loop gate involves a trio of residues that span both subunits (Figure 5F,G). In the GAT1587-mediated inhibited state, GIRK2-H233 has been shown to form cation–pi interactions with GIRK1-R313 at a bond length of ~3 angstroms (Figure 5G). GIRK1-R313 is a conserved residue in Kir channels and Gln and Trp variants in Kir1.1 have been identified to cause Bartter’s syndrome, a critical channelopathy in the kidney [28]. This interaction is relatively strong in the inhibited state and is broken through the switching mechanism of GIRK2-R230. In the ML297-mediated activated state, this interaction becomes over 4.5 angstroms. This frees GIRK1-R313 to salt bridge with GIRK2-E315 located on the GIRK2 G loop (Figure 5F,G). This results in the opening of the G-loop gate. In the GAT1587-inhibited channel state, GIRK2-E315 is free to salt bridge with GIRK1-K188, one of the critical basic lysines that was captured to form interactions with both the P4 and P5 phosphates of PIP_2_ in the pre-open state. This specific interaction of GIRK2-E315 with GIRK1-K188 had not been previously recognized, explaining the critical role of R313 in controlling the activity of both channel gates. These atomistic changes explain the differences seen in the transition from non-functional to functional states of the channel. ML297 and GAT1587 affect channel PIP_2_ interactions in opposite directions but equally for the GIRK1 and the partner subunit (the specific PIP_2_ is assigned a subunit based on where the basic lysines that bind that PIP_2_ originate from) (Appendix A). The effects that we see from ML297 on the GIRK1 subunit of either tetrameric channel usually range from a 1.1 to a 1.2-fold increase in the normalized salt bridge formation. For the partner subunit, the increase in the normalized salt bridge formation with PIP_2_ is on the order of 1.2-fold. For GAT1587, regardless of the subunit or heterotetramer, there is an approximate 0.8-fold decrease in the normalized salt bridge contacts (Appendix A). The very similar effects for these similar compounds further reinforced the idea that these compounds are not only nonspecific in their function, but also work through a similar mechanism to produce these effects. More broadly, these effects have been reproduced using multiple simulations. The initial sets of simulations provided an estimation of the size of the effects and the typical equilibrium for various collective variables in the channel closed and open states. From the original simulations, we conducted power analysis and were able to estimate that at least six replicates per simulation would be required for reliable statistical testing. These twenty-four additional simulations were conducted for additional validation of the differences between the induced channel states. These trajectories were analyzed identically to the original simulations. Channel gate distances over the last 5 ns of the simulations were used for statistical testing (Appendix A). The normalized salt-bridge formation was also pooled (Appendix A). Statistical testing using ANOVA revealed significant differences between our proposed inhibited and activated channel populations (Appendix A).

## 3. Discussion

Prior work provided evidence for in-silico gating of GIRK2 channels by Gβγ dimers and Na^+^ ions in the presence of PIP_2_, a finding that reproduced experimental results on the highly efficacious synergism of these gating co-factors [5,14]. The computational GIRK2 models also helped in validation of the binding site of urea-based activators and their allosteric control of channel–PIP_2_ interactions to open the channel [20]. The present study extends the computational models to heteromeric mammalian GIRK1-containing channels that have been resistant to purification and atomic resolution structure determination [7,8]. Here, we modeled GIRK1-containing channels (with GIRK2 and GIRK4) and utilized urea-based drugs (ML297 and the variant GAT1587) (16,17) that specifically bind GIRK1 and had been shown to switch from an activator to an inhibitor of heterotetramers through the addition of a small chemical moiety (a methyl cyclopropyl moiety). The models were successful in that in the presence of PIP_2_, ML297 activated GIRK1/2 and GIRK1/4 channels and GAT1587 inhibited them in a PIP_2_-dependent manner, in complete agreement with experimental results.

The approach of utilizing a minimally modified pair of oppositely acting ligands allowed us to identify not only specific differences in the binding of the two ligands but also profound changes in their mode of action. Our computational analysis focused on identifying differences between the two types of ligands and revealed three major and novel mechanistic points regarding Kir channel activation at large.
Kir channels (characterized by two transmembrane helices -TM1 and TM2- per subunit) utilize TM1 to relieve a restrain of the pore-lining TM2 (of the same subunit type) through the formation of an open-state-dependent TM1 hydrophobic wire (for GIRK1: F87, Y91, F87). These are absolutely or highly conserved residues within Kir channels (the Y91.1 position uses either Tyr or Phe). The ML297-induced hydrophobic wire is further stabilized by TM2 hydrophobic residues of the partner subunit (the other subunit type). This conformational state is communicated to the residue preceding the residue comprising the HBC gate (for GIRK1: M180) (Figure 6A).The TM1 hydrophobic wire residues couple to two acidic residues along the permeation pathway. The first, E141.1 at the C-terminal end of the Pore helix (PH), is conserved among GIRK channels (it is a Gln in every other Kir channel). The second, D173 in TM2, is a GIRK1-specific residue (among Kir3 channel subtypes, and Asp, Asn or Glu in every other Kir channel). Ion conduction requires that the PH residue is stabilized by the hydrophobic wire residue while the TM2 acidic residue needs to be freed from interactions with the hydrophobic wire residue (Figure 6A).The TM1 movement also repositions the Slide Helix (SH), a 10-aa helical structure arranged parallel to the inner leaflet of the plasma membrane that contains two critical acidic residues (D70 and D77). A CD loop basic residue interacts with the first SH acidic residue (D70) in the presence of the inhibitory ligand but switches to the second one (D77) when the stimulatory ligand binds and the SH moves. This new salt-bridge interaction positions a critical nearby His residue in the CD loop to let go of an important basic residue that controls both gates: the stimulatory ligand frees R313.1 from the GIRK2-H233 and allows to engage the GIRK2-E315 residue away from K188 of GIRK1 stabilizing the GIRK2 G-loop open gate indirectly and allowing the GIRK1-K188 to interact with PIP_2_, thus stabilizing its HBC open gate directly (Figure 6B).

A most surprising finding of this work was the link of the effects of the TM1 hydrophobic wire residues to the two TM2 acidic residues and the implications on ion conduction. As already mentioned, both the E141.1 (as E152.2D; ref. [26]) and the D173.1 (in enabling gating by ML297 and in its involvement in inward rectification) [19,27] have been implicated in ion permeation. Our study suggests that their coordinate regulation is critical and the question arises as to how. GIRK2-E152 has been shown computationally to aid with the process by which ions are conducted through the selectivity filter region [21]. Li and colleagues suggested that ion conduction requires a minimum number of ions to be present within the selectivity filter region and the channel for ion conduction to occur. Moreover, they recognized that the GIRK2-N184 was critical for efficient ion conduction, although the reason was not understood. The corresponding residue to GIRK2-N184 in GIRK1 is D173. We propose that there is a dipole created by E141 and D173 that is critical for the high efficiency of ion conduction seen in GIRK1-containing heteromers. When D173 is engaged in hydrogen-bonding interactions, and it is not free to attract K^+^ ions away from E141, then ions are trapped and conduction is decreased. In the GAT1587-mediated closed state of the channel, the GIRK1 Y91-D173 forms a very tight charge dipole hydrogen bond (2 angstroms apart). In comparison, the E141-W95 pair is over 7 angstroms apart. This configuration results in E141 being more negatively charged than D173 preventing ion conduction by trapping ions at the C-terminal end of the PH. Another important participant in the interactions with the Y91 and D173 residues of GIRK1 is the GIRK2-S181 residue, the residue next to the flexible G180 (corresponding to GIRK1-G169) where TM2 bends as the channel is gated open. The GIRK2-S181 interactions with Y91 and D173 of GIRK1 were through conventional hydrogen bonds ranging from 2 to 4 angstroms. Thus, the Y91, D173 of GIRK1 interactions with the S181 of GIRK2 not only limit the flexibility of G180 but also stabilize the GIRK1 Y91-D173 interaction creating the imbalance of the dipole in favor of GIRK1-E141 and trapping K^+^ ions away from the HBC channel gate. In conclusion, the employment of two very similar chemical probes that cause functionally meaningful and distinct conformational states allowed us to discover two major elements in channel gating: (1) the existence of state-induced TM1 hydrophobic wire and discern how it controls ion conduction through two key acidic residues in the permeation pathway and (2) a previously unrecognized interaction of a key basic residue (K188.1) destabilizing the G-loop gate when the channel is inhibited but switching to coordinate PIP_2_ and open the HBC gate in the presence of the activator.

## 4. Material and Methods

### 4.1. Generation of the GIRK1/X Homology Models

The GIRK2 crystal structure (PDBID:4KFM) is characterized as not fully opened (pre-open) and served as an initial template to model pre-open GIRK1/2 heterotetrameric ion channels. The Schrodinger 2019-1 package was used for model generation. The initial GIRK2 biological assembly was retrieved from the Orientation of Proteins in Membranes (OPM) into Maestro (Schrödinger, LLC, New York, NY, USA). The protein preparation wizard was used to assign bond orders, add hydrogens, create disulfide bonds, and fill in missing side chains using the Prime module. The initial structure included the Gβ and Gγ subunits. The missing side chains added in this process were Ile55, Arg73, Glu127, Phe141, Lys165, and Glu303 of GIRK2, Arg42 and Arg214 of the Gβ subunit, and Glu58 and Glu63, Phe67 of the Gγ subunit. Ionization and tautomeric states for heteroatom groups were generated using the Epik module at neutral pH (pH 7.0 ± 2.0) [29,30]. Protonated states of titratable residues were determined by p*Ka* calculations at physiological conditions (pH 7.0 ± 2.0) using the PROPKA module [29,30]. A restrained minimization was also performed on all atoms using the OPLS3 force field. The minimization was considered converged once heavy atom displacement was below 0.3 angstroms.

To address the lower sequence homology between GIRK1 and GIRK2, we developed a chimeric homology model utilizing the cytosolic domain of GIRK1 that had been previously crystalized (PDBID:1U4E) and the GIRK2 crystal structure (PDBID:4KFM). The cytoplasmic GIRK1 domain was prepared identically to the preopen GIRK2 structure. A protein structural alignment was used to superimpose the two crystal structures. Homology modeling was performed using the structure prediction wizard included with the Prime module. For modeling of the GIRK1, the canonical sequence (Primary accession number: P48549) was used. For modeling of GIRK2 and GIRK4, the canonical sequences (GIRK2 Primary accession number: P48542, GIRK4 Primary accession number: P48544) were used.

These three sequences were aligned using ClustalW (see Appendix A for sequence and structure alignments used for model generation). GIRK2-P198 was selected as the endpoint for the transmembrane domain of the pre-open GIRK2 crystal structure [31]. Ramachandran outliers were included for the GIRK1/2 (Appendix A) and GIRK1/4 (Appendix A) heterotetramers. For each model, there were fewer than four residues with side chains outside the allowed region, attesting to the goodness of the homology models. The DiC1-PIP_2_ was included as a co-factor when building each of the subunits. DiC1-PIP_2_ and Na^+^ were used as co-factors when building GIRK2/4 subunits. Only DiC1-PIP_2_ was used for building GIRK1 subunits. The four individual subunits of the heterotetrameric ion channels were built simultaneously. C4-symmetry was implicitly considered for the homology model building as each subunit to be built was pre-aligned to a specific chain. The resulting homology models were used in docking studies.

PIP_2_ was built based on the DiC1-PIP_2_ that is found within the GIRK2 homotetrameric biological assembly. The alkyl tails of PIP_2_ were manually built onto the DiC1-PIP_2_ co-factor. The resulting alkyl tails were then selected for minimization while the phospho- headgroup and protein were held fixed. The minimization was considered converged once heavy atom displacement was below 0.3 angstroms.

### 4.2. Ligand and Co-Factor Parameterization

Ligands were built in Maestro. Ligands were then imported into GaussView 6.1 [32]. To generate same initial configurations for each ligand for geometry optimization, initial semi-empirical geometry optimization calculations at the PM6 level were carried out using Gaussian16 rev B0.1. These were then followed by geometry optimizations carried out at the B3LYP/cc-pVTZ, M06/cc-pVTZ, and wB97XD/cc-pVTZ levels. Optimized structures were used as the initial coordinates for electrostatic potential charge (ESP) generation utilizing the same functional (B3LYP, M06, or wB97XD) as in the previous optimization step. While there was good agreement between all 3 functionals, final RESP charges from those used in the all-atom molecular dynamics systems were drawn from the structures treated at the M06/cc-pVTZ level. RESP charges were generated using the antechamber module of AmberTools18 based on derived ESP charges and the GAFF2 force field.

PIP_2_ was built based on the DiC1-PIP_2_ that is found within the GIRK2 homotetrameric biological assembly. The alkyl tails of PIP_2_ were manually built onto the DiC1-PIP_2_ co-factor. The resulting alkyl tails were then selected for minimization while the protein was held fixed. This minimization was performed using the OPLS3 force field and the Prime module. A PIP_2_ molecule was then extracted from this system for parameterization. Gaussian09 was used for structure optimization and charge generation. This PIP_2_ structure was optimized at the Hartree–Fock/6–31G* level. ESP charges were generated at the same level. RESP charges were generated using the antechamber module of Amber Tools 17. Final force field parameters were derived from these RESP charges and the GAFF2 force field [33,34].

### 4.3. Ligand Docking and Model System Generation

The energy-based homology models of the pre-open GIRK1/X heterotetrameric ion channels were used in follow-up docking calculations. The co-factors used to build these models were removed before induced-fit docking. The induced-fit docking protocol used is in Schrodinger 2019-1 [35]. Based on previous studies, the initial binding site was defined based on GIRK1 residues Phe97 and Phe175. Ligands that had been parameterized quantum mechanically were used in the docking studies. The normal induced-fit protocol was modified. In the normal workflow, typically 20 protein-ligand complexes would go onto the refinement stage. To better sample the possible ligand conformations, 80 complexes were instead advanced to the refinement stage, similar to the extended sampling induced-fit docking protocol. Non-planar conformations for amides were penalized. Due to the pi conjugation of the GAT1587 and ML297 compounds, the torsional potential around the core urea was increased to penalize non-planar conformations. An implicit membrane was used in the protein refinement stage. This membrane was defined by the GIRK2 OPM structure. The final redocking stage used Glide XP for docking, post-docking minimization, and scoring. This process was repeated for each GIRK1 subunit. The best-scoring poses for each compound were combined. This resulted in two ligands per system with each GIRK1 being occupied.

Using CHARMM-GUI, the heteromeric channel complexes were immersed in explicit lipid bilayer of 1-palmitoyl-2-oleoyl-sn-glycero-3-phosphocholine, 1-palmitoyl-2oleoyl-sn-glycero-3-phosphatidylethanolamine, 1-palmitoyl-2-oleoyl-sn-glycero-3phosphatidylserine, and cholesterol with molecular ratio of 25:5:5:1 [36]. An initial guess of 100 angstroms was used to define the size of the simulation cells. This defined approximately 14 Å from the protein periphery in each dimension using periodic boundary conditions. 150 mm KCl was added to the systems. TIP3P was used as the water model for the systems. Parameters for ions were derived from the TIP3P leap rc file and included a 12-6 non-bonded correction. This file is distributed along with Amber Tools. TIP3P is the recommended water model for use with FF14SB force field and includes additional parameters for ions. The FF14SB, LIPID17, and GAFF2 force fields were used for protein, mixed lipid membrane, and PIP_2_, respectively. The systems were approximately 140,000 atoms each.

### 4.4. All-Atom MD Simulations

All molecular dynamics simulations were performed in Amber 18 using reference GeForce GTX 1080 or Tesla P100 GPUs [37,38]. All simulations were performed in triplicate. A two-stage minimization protocol was used that combined the steepest descent algorithm (10,000 steps) and conjugate gradient (10,000 steps) for each model system. The systems were heated from 0 to 300 K using the Langevin thermostat algorithm with a 1-fs time step to avert internal disturbance. The protein and lipid bilayer were initially fixed to remove any potential steric clashes from K^+^ or Cl^−^ ions, and water molecules; followed by gradually reduced position restraints on the protein and membrane (10 to 0.1 kcal/mol·Å^2^ in 6 steps of total 3 ns).

A 7 ns MD run was conducted using the constant-temperature, constant-pressure ensemble (NPT) without an electric field to equilibrize the systems after restraint release. This was done using a 2 fs time step. These simulations were followed with a 0.3-μs MD simulation under a constant-temperature, constant-volume ensemble (NVT) with an applied electric field. To induce ion permeation in the time scale of simulations, an external voltage of −0.06 V/nm [13,39] was employed. The PMEMD.CUDA program in AMBER18 was used to conduct all simulations. Long-range electrostatics were calculated using the particle mesh Ewald method with a 10-Å cutoff. A 3-fs time step by employing a hydrogen mass repartition algorithm for system solutes [40] was used to accelerate the MD simulations. The SHAKE algorithm was used to treat the solvent molecules. The constant voltage is generated through an applied electric field using the efz keyword on the input parameters. This keyword specifically defines an electric potential along the *z*-axis of the system. Given that we pre-aligned each model, the orientation of the 4KFM protein structure in the membrane, moving from positive z to negative z represents going from the extracellular side of the plasma membrane, through the plasma membrane, and entering the intracellular side of the plasma membrane. A negative electrostatic potential would drive positively charged ions from positive z to negative z. Finally, an electric field of 0.06 kcal/(mol*A*e) corresponds to approximately 210 mV across the 35 angstrom plasma membrane, which approximates a physiological range of membrane potentials when the channel opens. We used an NVT ensemble for our MD runs as required in AMBER when using an applied electric field.

### 4.5. Analysis of MD Simulations

All analyses were performed on the trajectories utilizing an external electric field of 0.06 V/nm. After removal of water, membrane lipids, and negatively charged ions, production trajectories were converted to dcd format for follow-up analysis. The program Simulaid was used to generate the normalized salt bridge formation [41]. The default upper cut-off for salt bridge formation was used. Salt bridges were defined based on the atom names. The atom name “NZ” was used to define the terminal nitrogen of basic lysine groups. The atom names “O11, O12, O13, O14, O15, O16, P2, P3” defined the phosphate groups of the PIP_2_ molecules. The default cut-off for all distance measurements was taken using CPPTRAJ over the last 250 ns of the simulations [42]. These distances were represented as histograms using SPSS.

The Bio3D package was used to perform a community network analysis utilizing only the alpha carbons of the protein [43]. While this community network analysis was not used in this study, the underlying dynamic cross-correlation matrix produced in that workflow served as the input for the shortest pathway analysis to interrogate correlated residues of the protein [44]. Then, 250 suboptimal pathways were generated using this workflow. Finally, these were visualized in VMD.

### 4.6. Shortest Pathway Analysis

The Bio3D package was used to perform a correlation analysis throughout the last 150 ns of the 300 ns trajectories [43]. For this analysis, the molecular dynamics trajectories were stripped of water and membrane molecules. Using the amber topology file, c alpha atoms were used to define the position of each residue. This working trajectory was then used to define a dynamic cross-correlation network using the dccm function of Bio3D. Community network analysis was also performed for these trajectories using the cna function with a cutoff cij of 0.4. All other options for the cna function were left as the default values. This community network analysis was then used as the input for the can path function to perform the suboptimal pathway analysis for the correlated network [44]. For each source/sink pair, 250 pathways were generated. These pathways were then visualized in a spline form using VMD. The weighting of each spline indicates the number of pathways that are transferred through a specific residue pair. The GIRK1 residue F97 served as the source for each pathway. The sink residue was defined as a key functional residue that is part of either the CD loop or the PIP_2_ binding site.

### 4.7. Chemical Synthesis

Chemicals and synthesis of ML297 and its derivative GAT1587 were as recently described in [20].

### 4.8. Electrophysiology

*Xenopus laevis* oocyte expression: Plasmid DNAs of GIRK channel subunits were linearized before in-vitro transcription. Capped RNAs were transcribed using mMESSAGE mMACHINE T7 Transcription Kit (Thermo Fisher Scientific, Waltham, MA, USA). *Xenopus oocytes* were surgically extracted, dissociated, and defolliculated by collagenase treatment, and microinjected with 50 nL of a water solution containing 1 ng of each GIRK subunit RNA. For TEVC experiments, oocytes were kept 2 days at 17 °C before recording. Two-electrode voltage-clamp and data analysis: Whole-oocyte currents were measured by two-electrode voltage clamp (TEVC) with GeneClamp 500 (Molecular Devices, San Jose, CA, USA), or TEC-03X (NPI) amplifiers. Electrodes were pulled using a Flaming-Brown micropipette puller (Sutter Instruments, Novato, CA, USA) and were filled with 3 M KCl in 1.5% (*w/v*) agarose to give resistances between 0.5 and 1.0 MΩs. The oocytes were bathed in ND96 recording solution comprising, in mM: KCl 2, NaCl 96, MgCl_2_ 1 and HEPES 5, buffered to pH 7.4 with KOH. Where indicated, GIRK channel currents were assessed in a high K^+^ recording solution comprising, in mM: KCl 96, NaCl 2, MgCl_2_ 1 and HEPES 5, buffered to pH 7.4 with KOH. Currents were digitized using a USB interface (National Instruments, Austin, TX, USA) and recorded using WinWCP software (University of Strathclyde, Glasgow, UK). To study GIRK channels, oocytes were held at 0 mV, and currents were assessed by 100 ms ramps from −80 to +80 mV that were repeated every second. The effect of the reagents was determined at −80 mV, then the channels were blocked by 5 mM BaCl_2_. Block was expressed as the percent-current block normalized to the maximum current. At least 6 oocytes from at least two different *Xenopus* frogs were studied per experiment.

## Figures and Tables

**Figure 1 ijms-23-10820-f001:**
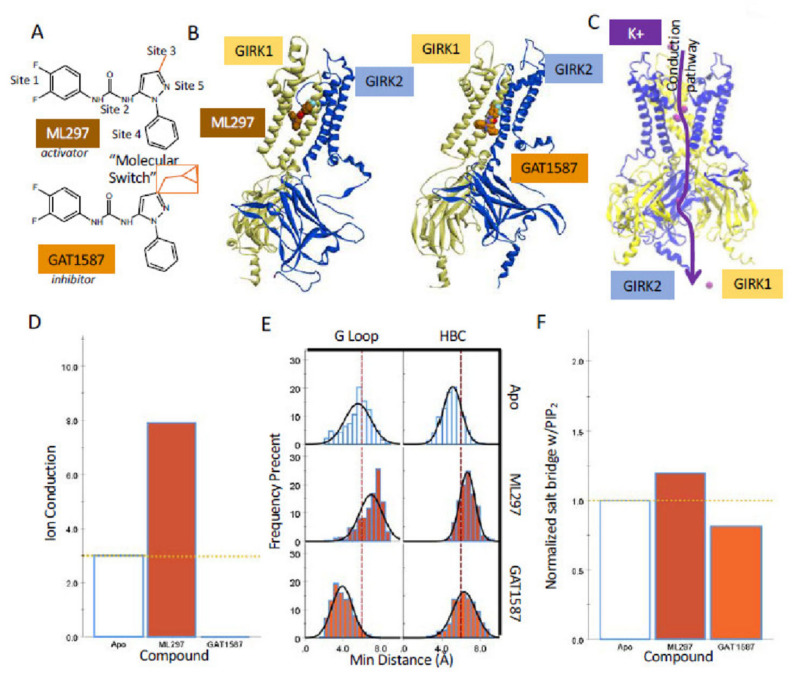
Pharmacological actions of a molecular switch moiety are reproduced in sub-microsecond MD simulations in GIRK1/2. (**A**) The structures of GIRK1 specific modulators ML297, GAT1587, and the site of molecular switching region (off the pyrazole ring) that controls compound activity. (**B**) Equilibrium binding site for ML297 and GAT1587 following a 300 ns stochastic dynamics simulation in complex with PIP_2_ and GIRK1/2. (**C**) Ion permeation pathway for a single ion taken from the ML297-PIP2-GIRK1/2 simulation. (**D**–**F**) Ions conducted, minimum gate distances, and normalized salt-bridge formation over the last 150 ns of the simulations are shown.

**Figure 2 ijms-23-10820-f002:**
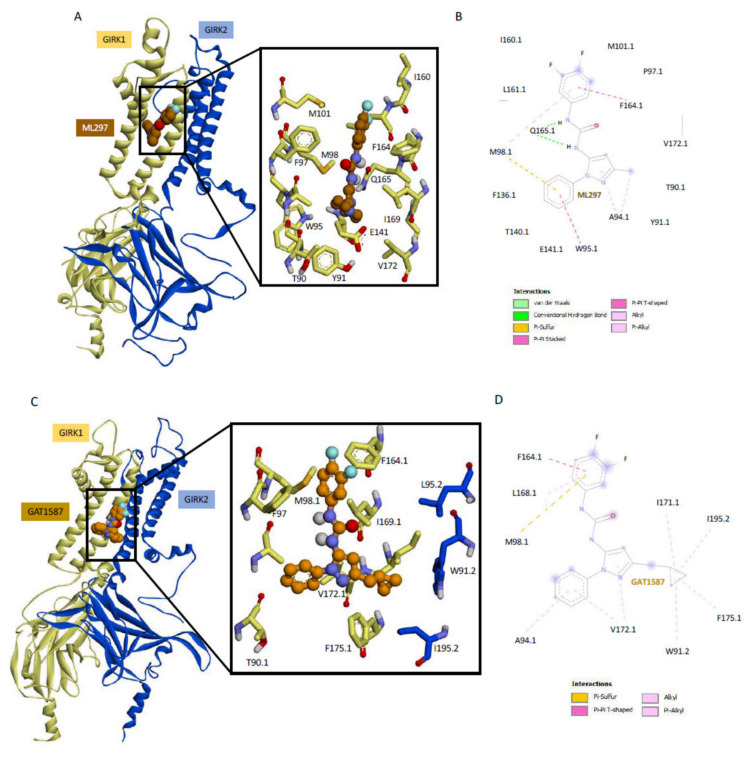
GAT1587 binding site along the TM helices in the GIRK1/2 heterotetramer.(**A**) Equilibrium binding pose for ML297 when complexed with PIP_2_ GIRK1/2 after 300 ns of stochastic dynamics. (**B**) A 2D schematic representation of the compound’s binding pose depicting protein ligand interactions. (**C**) Equilibrium binding pose for GAT1587 when complexed with PIP_2_ GIRK1/2 after 300 ns of stochastic dynamics. (**D**) A 2D schematic representation of the compounds binding pose depicting protein ligand interactions.

**Figure 3 ijms-23-10820-f003:**
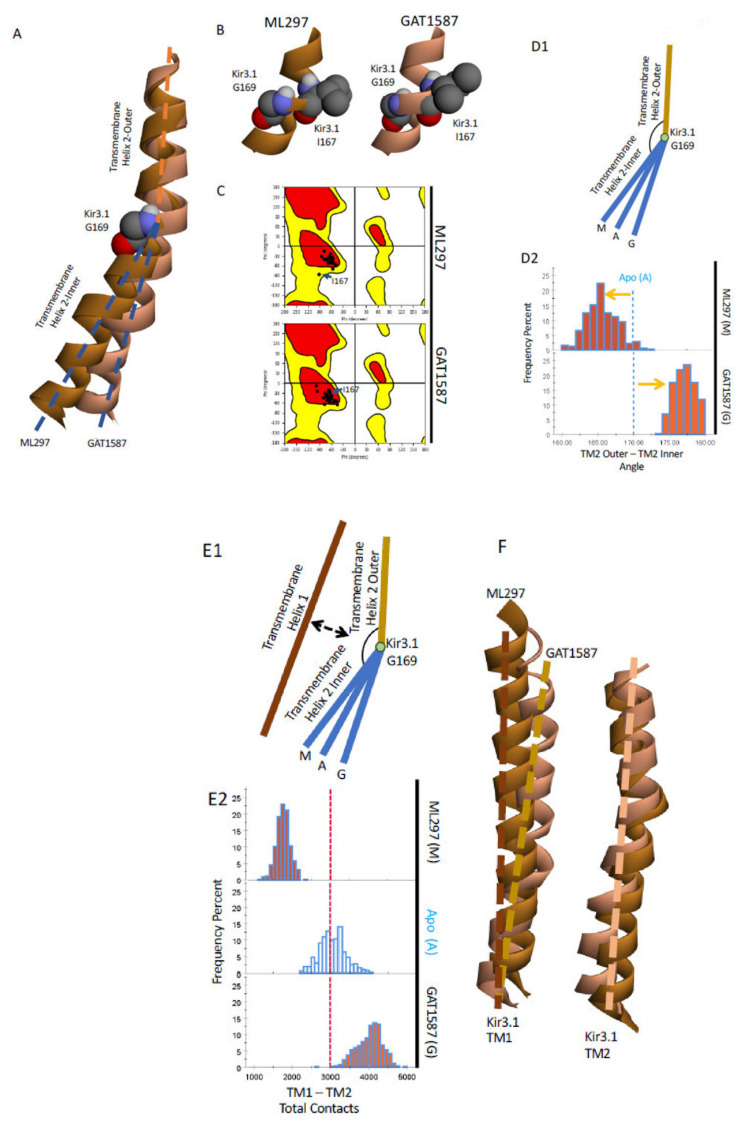
ML297 decreases and GAT1587 increases the angle between the outer and inner TM2 segments on each side of the flexible GIRK1-G169. (**A**) Comparison between TM2 of the ML297 (dark orange) and the GAT1587 (light orange) complexed with the GIRK1/2 systems. (**B**) The two hinge points that allow for TM2 bending. (**C**) Ramachandran plots for residues within TM2. I167 is highlighted in both. (**D**) Schematic representation of the movements of TM2 and a quantification of the bending around GIRK1-G169. (**E**) Expanded model including the nearby TM1 and a quantification of these effects. (**F**) Relative movements of the two transmembrane helices.

**Figure 4 ijms-23-10820-f004:**
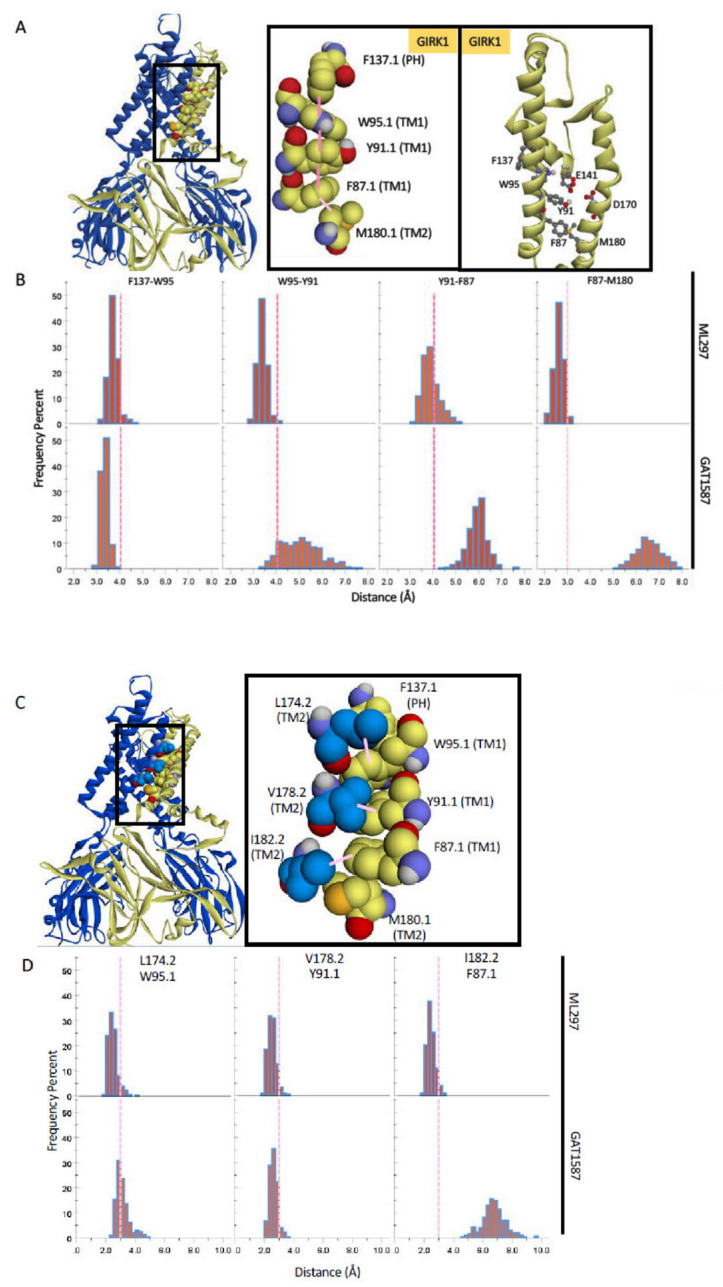
A hydrophobic wire (W95-Y91-F87) couples TM1 to TM2 at the level of the HBC gate (M180) and is stabilized by ML297 but not by GAT1587 to regulate the E141/D173-dependent conduction. (**A**) Protein structure with the pi-stack along TM1 highlighted. (**B**) Interaction distributions in the presence of ML297 or GAT1587. Pink denotes pi–pi interactions with an upper cutoff of 4 angstroms. Light pink denotes Van der Waals interactions with an upper cutoff of 3 angstroms. (**C**) Protein structure with the stabilizing hydrophobic residues of the GIRK2 subunit that can interact with the TM1 hydrophobic wire residues. (**D**) Interaction distance distributions of hydrophobic residues involved when either ML297 or GAT1587 is present. Light pink denotes Van der Waals interactions with an upper cutoff of 3 angstroms. (**E**) Protein structure (**E1**) with acidic residues that interact with a potassium ion (highlighted interactions) (**E2**). Dotted lines indicate a broken interaction between residues. The solid lines indicate the formation of an interaction. (**F**) Interaction distance distributions of TM1 hydrophobic chain residues with the two residues of the dipole between D141 and D173 of GIRK1 and S181 (next to G180) of GIRK2 in the presence of ML297 or GAT1587. Light green denotes dipole integrations with an upper cutoff of 4 angstroms. (**G**) Schematic representation of the charge relay network between W95 and Y91 of the TM1 hydrophobic chain with the two acidic residues.

**Figure 5 ijms-23-10820-f005:**
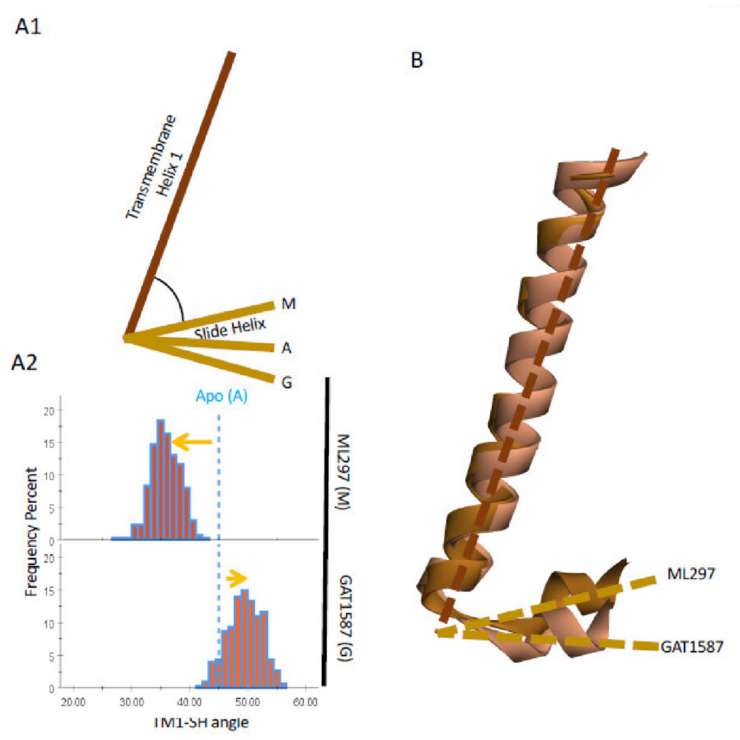
The ML297-induced SH movement, as a result of the TM1 movement, drives changes in the CD loop interactions causing stabilization of the G-loop in the open conformation through its residue GIRK2-E315.2 liberating K188.1 that coordinates PIP_2_ to stabilize the HBC gate in the open conformation. (**A**) A model of how TM1 modulates the slide helix (**A1**) and a quantification of these effects (**A2**). (**B**) Relative movements of the activated and inhibited SH regions. (**C**) Protein structure (**C1**) with residues that link the SH to the CD loop (**C2**). (**D**) Distance distributions of key residues that drive the channel into the active state. Dark yellow denotes charge–charge interactions with an upper cutoff of 4.0 angstroms. Light green denotes dipole charge interactions with an upper cut off of 4.0 angstroms. (**E**) Schematic outline of the changes in key residue interactions. (**F**) Protein structure (**F1**) with residues that link the SH to the CD loop (**F2**). (**G**) Distance distributions of residues that drive the channel into the active state. Dark yellow denotes charge–charge interactions with an upper cutoff of 4.0 angstroms. Pink denotes dipole charge interactions with an upper cutoff of 4.0 angstroms.

**Figure 6 ijms-23-10820-f006:**
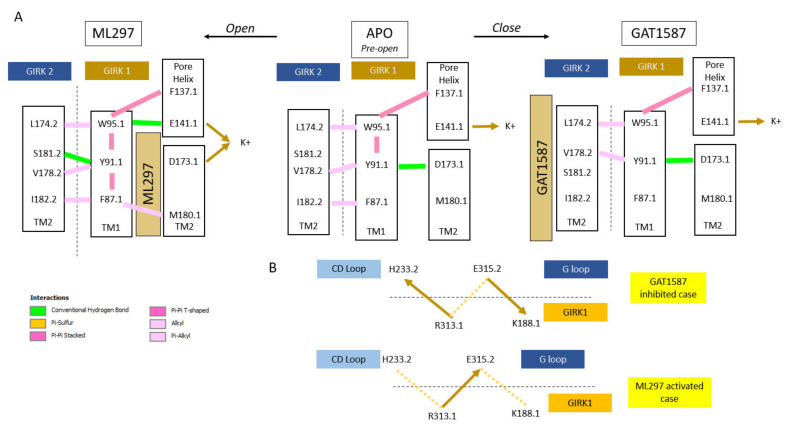
Overview of residue interactions driving the pre-open Apo channel state to the ML297-induced open conformation versus the GAT1587-induced close conformation. (**A**) A summary of the interaction networks that control channel function located near the compound binding site. (**B**) Outline of key changes affecting G loop open-state stabilization or a key Lys that coordinates PIP_2_ to open the HBC gate.

## Data Availability

DOI: https://doi.org/10.5281/zenodo.7083203, accessed on 20 March 2022.

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
