# Peer review of "Use of a Molecular Switch Probe to Activate or Inhibit GIRK1 Heteromers In Silico Reveals a Novel Gating Mechanism"

_ijms, 2022, doi:10.3390/ijms231810820_

Round 1
Reviewer 1 Report
The paper by Gazgalis et al. analyzes the molecular mechanisms of regulation of GIRK1-containing heterotetramers, GIRK1/2 and GIRK1/4, by a urea-based opener ML297 and a blocker GAT1587 using molecular docking and molecular dynamics. The two molecules differ only in one position (the molecular switch region), yet they have opposite effects on gating. The paper offers a detailed insight into drug binding sites and drug-induced changes in channel gating on atomistic level. The modeling also reveals several new important mechanisms that may pertain to the general gating mechanism of GIRK channels, in particular GIRK1/x heterotetramers. These include the formation (and potential general importance in gating) of an ML297-induced hydrophobic “wire” that drives the conformational change from the drug binding sites in the transmembrane regions of GIRK1 to the gates and their link to the permeation pathway, and the roles of several conserved amino acids that participate in channel gating. Overall, it is an interesting and important paper that may add significantly to our understanding of gating and pharmacology of GIRK channels.
Major comments:
- Some of the methods used, and some of the validation procedures, are not fully explained or presented.
1a. The homology modeling (HM; lines 592-628) of GIRK1/2 and GIRK1/4 and can be considered as a significant step toward better modeling of GIRKs. However, it is not trivial, and it is important to strengthen the validity of the HM models and simulations based on them. Validation can be strengthened in several ways. I suggest to show sequence alignment (clustalW), which was used to build the homology models (maybe in Supplemental Material). Further, depending on the possibilities of the programs used for HM construction, indicators of goodness of HM could for example be sidechain outliers, clash scores, Ramachandran plots, or local confidence scores. Eventually, the PDBs of the HM-derived structures should be placed in a public repository.
Lines 653-654 state: “The cofactors used to build these models were removed prior to induced-fit docking”. What co-factors? Presumably Gβγ molecules present in 4kfm; anything else? Please detail.
1b. Line 34-35/655, Docking: Local docking of both compounds near resides F79 & F175 was conducted to obtain starting conformations for the MD simulation (lines 34-35, 655). It would be important to specify why this site is selected, or provide a citation. Further, it will helpful to provide the docking poses used for starting the simulations, and indicate how many ligand molecules have been placed in the simulation volume, per heterotetramer (1, 2 or more). Also, have you tried MD simulations without prior pre-docking of the drugs?
1c. Lines 132-145, 166-168: is 300 ns MD simulation sufficient to investigate binding & gating effect of activator/inhibitor? To better demonstrate this, some validation for the equilibration of the system over the last 150 ns can be shown (e.g. RMSD of the ML297/GAT1587 over the last 150 ns of the runs). To show that the protein conformation is equilibrated, one can use RMSD of the backbone/C-alphas of the heterotetramers over the last 150 ns of the runs.
1d. Lines 140, 109, 678-696: It is mentioned that stochastic simulations were conducted. Are they different from standard (deterministic) MD simulations? In this case, please provide a description in the Methods. It would be helpful to have a table (in the supplement) giving an overview over all simulations conducted, with the relevant conditions indicated.
1e. Lines 292-294: Please explain how analysis of the restraining potential was conducted.
- The paper needs a thorough proofreading; there are several issues with grammar and many typos. Some of them are listed in the following.
Suggestions for improvement:
Suggestions for improvement of description and presentation of results.
The paper contains a large number of important details (e.g. amino acid residue numbers) within sentences, which sometimes impedes smooth reading. Also, the explanations in legends and captions in the figures are sometimes too brief or missing. I would like to make several suggestions to enhance the clarity and improve readability.
- Lines 147-150: It would be helpful to describe the way the different production runs were treated during analysis (for distance measurements, conduction events, bending of TM, etc.), i.e. how the collective variables were obtained (mean over all runs?). It might be advisable to provide the selective variables for each of the replica runs to see the variability among them.
- Throughout the paper, the numbers of various a.a. residues that play important roles in the described phenomena, are listed for all 3 subunits used. It might be time- and space-saving, and also helpful for better orientation of the reader, to make a list or a table of the mentioned residues in all 3 subunits.
Suggestions for improvement/corrections of Figures.
- Fig. SF1 is very useful for the reader. It might further assist in understanding the results of the study better if the authors add/label channel’s domains mentioned in the text in its different parts (e.g. lines 62-64): TM1, TM2, NTD, CTD, and the CD loop.
- In Fig. SF1D, the captions are not clear (too small). The quality of the upper panel should be improved.
- Figs. 1, SF1, SF4: Please explain the units of ion conduction and include the definition of a normalized salt-bridge. (Please define in Methods as well). Lines 164-165 state: “we…monitor the number of ions conducted through the channel over the course of each simulation”; is it the number of ions that completely traverse the channel, or SF? In what time?).
- Figure 1E: the units of the distance should be provided.
- Legends to figures 2B and 2D: “A 2D schematic representation of the compound’s binding pose depicting protein ligand interactions and solvent accessibility.” The figure only shows interactions.
- Figure 2: Legend is wrong, A and B should be switched with C and D. Fig. 2C: Not all residues are labeled in the right figure.
- Figure 4 B, D: The units of axis labels are missing. Especially the unit for the distance measurements should be provided. A, C: Showing residues in stick representation, and showing the HBC gate, might help to increase the clarity of the figure.
- Fig. 4 E, G: a) W95 is not in the scheme. b) Please define the meaning of arrows and dotted lines. (This will also be helpful for other figures).
Typos and grammar
- Line 103: Change “inhibited to and activated” to “inhibited to an activated”
- Section 2.2 (line 202): consider changing the title of the section, it mainly discusses conformations rather than PIP2 interactions.
- Lines 204, 219. etc.: please explain what is meant by nonspecific nature of ML297 and GAT1587?
- Lines 230-232: “Notably, the methyl cyclopropyl ring interacts with GIRK4-W86 and not GIRK4-I190.” I190 is not labelled in the corresponding figure (Fig 2C, D). Should I195 be written there?
- Line 231: change confirmations to conformations.
- Section 2.4 title is long and complicated. Consider rephrasing.
- Please check typos in lines 399-402; 410-416 (typos, commas missing, sentence about D173 is unclear); 473-484; line 526 (what do you mean by “These two transmembrane (TM) per subunit channels”?
- Section 2.5 title is very long and convoluted.
- Line 724: change from to form.
Author Response
REVIEWER 1
The paper by Gazgalis et al. analyzes the molecular mechanisms of regulation of GIRK1-containing heterotetramers, GIRK1/2 and GIRK1/4, by a urea-based opener ML297 and a blocker GAT1587 using molecular docking and molecular dynamics. The two molecules differ only in one position (the molecular switch region), yet they have opposite effects on gating. The paper offers a detailed insight into drug binding sites and drug-induced changes in channel gating on atomistic level. The modeling also reveals several new important mechanisms that may pertain to the general gating mechanism of GIRK channels, in particular GIRK1/x heterotetramers. These include the formation (and potential general importance in gating) of an ML297-induced hydrophobic “wire” that drives the conformational change from the drug binding sites in the transmembrane regions of GIRK1 to the gates and their link to the permeation pathway, and the roles of several conserved amino acids that participate in channel gating. Overall, it is an interesting and important paper that may add significantly to our understanding of gating and pharmacology of GIRK channels.
We thank the reviewer for the overall positive review. Our responses to the constructive critique appear below in blue font and corrections to the revised manuscript have been made indicating the exact lines where they can be found. We have also tracked changes over the original manuscript so that all changes made can be readily identified.
Major comments:
- Some of the methods used, and some of the validation procedures, are not fully explained or presented.
1a. The homology modeling (HM; lines 592-628) of GIRK1/2 and GIRK1/4 and can be considered as a significant step toward better modeling of GIRKs. However, it is not trivial, and it is important to strengthen the validity of the HM models and simulations based on them. Validation can be strengthened in several ways. I suggest to show sequence alignment (clustalW), which was used to build the homology models (maybe in Supplemental Material). Further, depending on the possibilities of the programs used for HM construction, indicators of goodness of HM could for example be sidechain outliers, clash scores, Ramachandran plots, or local confidence scores. Eventually, the PDBs of the HM-derived structures should be placed in a public repository.
1a) A chimeric model was used for both the GIRK1/2 and GIRK1/4 models. This has been clarified in the methods section. In addition, we show the sequence alignment used to build the homology models (new figure SF 9B) and list the percent identity (SF 9C) and homology (SF 9D) in the supplemental figures. Ramachandran outliers have also been included for the GIRK1/2 (SF 10A) and GIRK1/4 (SF 10B). For each model, there are less than 4 residues with side chains outside the allowed region. The inclusion of these figures can be found in the manuscript [lines 643-648].
Lines 653-654 state: “The cofactors used to build these models were removed prior to induced-fit docking”. What co-factors? Presumably Gβγ molecules present in 4kfm; anything else? Please detail.
By “co-factors” we mean anything else besides GIRK subunits needed to gate the channel optimally (namely, PIP2, Gbg and Na+, now clarified in the revised manuscript – lines 66-67). For GIRK1 subunits, we modeled the channels with the DiC1 PIP2 and the Gbg from the 4KFM structure. The GIRK2 and GIRK4 subunits that can also bind Na+ were modeled with the DiC1 PIP2, the Gbg, and Na+ ions of the 4KFM structure.
We plan to use the server shown below to share our model structures. https://zenodo.org/1b. Line 34-35/655, Docking: Local docking of both compounds near residues F97 & F175 was conducted to obtain starting conformations for the MD simulation (lines 34-35, 655). It would be important to specify why this site is selected, or provide a citation. Further, it will helpful to provide the docking poses used for starting the simulations, and indicate how many ligand molecules have been placed in the simulation volume, per heterotetramer (1, 2 or more). Also, have you tried MD simulations without prior pre-docking of the drugs?
These residues were shown to be required for ML297and GAT1508 activity [in our work, ref. 20]. In our previous paper detailing the allosteric effects of GAT1508, we used these residues to define the binding site [lines 149-156 of revised manuscript] and saw an increase in the normalized PIP2 channel salt bridge formation. In a similar manner to the previous work, two compounds were included per heterotetramer, both being docked to the GIRK1 subunits. We have not tried simulations without pre-docked compounds. Initial binding poses of ML297 and GAT1587 bound to GIRK1/2 and GIRK1/4 are shown in a new Supplemental Figure (Fig. SF 3) and mentioned in the revised manuscript [lines 236-238].
1c. Lines 132-145, 166-168: is 300 ns MD simulation sufficient to investigate binding & gating effect of activator/inhibitor? To better demonstrate this, some validation for the equilibration of the system over the last 150 ns can be shown (e.g., RMSD of the ML297/GAT1587 over the last 150 ns of the runs). To show that the protein conformation is equilibrated, one can use RMSD of the backbone/C-alphas of the heterotetramers over the last 150 ns of the runs.
Six replicas of 300 ns MD runs for GIRK1/2 and GIRK1/4 each with either the activator (ML297) or the inhibitor (GAT1587) are now shown to assess changes in RMSD as a function of the simulation time (Fig. SF 4, SF 5). It can readily be seen that within 50 ns into the simulation the systems equilibrate. This is now mentioned in the revised manuscript [lines 239-242].
1d. Lines 140, 109, 678-696: It is mentioned that stochastic simulations were conducted. Are they different from standard (deterministic) MD simulations? In this case, please provide a description in the Methods. It would be helpful to have a table (in the supplement) giving an overview of all simulations conducted, with the relevant conditions indicated.
Colloquially, molecular dynamics is used as a catch-all term for different sets of deterministic and nondeterministic simulations. Molecular dynamics, however, specifically refers to the microcanonical ensemble in which the number of particles (N), the volume (V), and the total energy of the simulation (E) are held constant (i.e., an NVE ensemble). Langevin or stochastic dynamics through the introduction of random variables generates a canonical ensemble by which the temperature of the system is held constant (i.e., an NVT ensemble). Most commonly, simulations are done as an isothermal–isobaric ensemble where the pressure (P) is also held constant (i.e., NPT ensemble). Both the canonical and isothermal–isobaric ensembles are more representative of the thermodynamic conditions typically present in experiments. However, when using an applied electric field in AMBER, a canonical ensemble must be used, which is what we have done. This has now been indicated in the Methods section of the revised manuscript [lines 736-746].
1e. Lines 292-294: Please explain how analysis of the restraining potential was conducted.
The existence of a naturally occurring restraining potential was suggested through the number of contacts made between the two transmembrane helixes. No further analysis was conducted.
Suggestions for improvement of description and presentation of results.
The paper contains a large number of important details (e.g., amino acid residue numbers) within sentences, which sometimes impedes smooth reading. Also, the explanations in legends and captions in the figures are sometimes too brief or missing. I would like to make several suggestions to enhance the clarity and improve readability.
- Lines 147-150: It would be helpful to describe the way the different production runs were treated during analysis (for distance measurements, conduction events, bending of TM, etc.), i.e. how the collective variables were obtained (mean over all runs?). It might be advisable to provide the selective variables for each of the replica runs to see the variability among them.
We have now analyzed statistically the results from different replica runs (Figs, SF 4 and 5 and statistical analysis SF 11 and 12 and Table ST 1) to provide measures of variability and statistical significance of differences we report in our chosen collective variables. This has been clarified in the revised manuscript [lines 172-175, 517-528].
- Throughout the paper, the numbers of various a.a. residues that play important roles in the described phenomena, are listed for all 3 subunits used. It might be time- and space-saving, and also helpful for better orientation of the reader, to make a list or a table of the mentioned residues in all 3 subunits.
We have now included a sequence alignment of the 3 subunits used (GIRK1, GIRK2, and GIRK4) as well as the truncated constructs for which atomic resolution structures have been determined that served as our starting points for homology modeling and simulations (Fig. SF 9). Residues have been identified by numbers so the readers should now be able to find all residues mentioned and compare with the other channel subunits. In the text this is addressed in lines 643-645.
Figures
- SF1 is very useful for the reader. It might further assist in understanding the results of the study better if the authors add/label channel’s domains mentioned in the text in its different parts (e.g. lines 62-64): TM1, TM2, NTD, CTD, and the CD loop.
This has been addressed as the SF1 was split into 2 figures for clarity. Requested labels have been added.
- In Fig. SF1D, the captions are not clear (too small). The quality of the upper panel should be improved.
This has been addressed as the SF1 was split into 2 figures for clarity.
- 1, SF1, SF4: Please explain the units of ion conduction and include the definition of a normalized salt-bridge. (Please define in Methods as well). Lines 164-165 state: “we…monitor the number of ions conducted through the channel over the course of each simulation”; is it the number of ions that completely traverse the channel, or SF? In what time?).
Ion conduction was defined as a single ion traversing both sets of gates, the G loop, and helix bundle crossing gates, through the last 150ns of the simulation [lines 190-193]. We did also observe passages through the selectivity filter as well.
- Figure 1E: the units of the distance should be provided.
The unit of distance is provided (Å or Angstroms)
- Legends to figures 2B and 2D: “A 2D schematic representation of the compound’s binding pose depicting protein ligand interactions and solvent accessibility.” The figure only shows interactions.
Solvent accessibility has been removed.
- Figure 2: Legend is wrong, A and B should be switched with C and D. Fig. 2C: Not all residues are labeled in the right figure.
Corrected!
- Figure 4 B, D: The units of axis labels are missing. Especially the unit for the distance measurements should be provided. A, C: Showing residues in stick representation, and showing the HBC gate, might help to increase the clarity of the figure.
The unit is in Angstroms (Å)
- 4 E, G: a) W95 is not in the scheme. b) Please define the meaning of arrows and dotted lines. (This will also be helpful for other figures).
W95 was added to the summary diagrams. The dotted lines indicated a loss of an interaction. The arrows indicate the gain of an interaction. This was added to the caption for Figure 4.
Typos and grammar
- Line 103: Change “inhibited to and activated” to “inhibited to an activated”
Corrected!
- Section 2.2 (line 202): consider changing the title of the section, it mainly discusses conformations rather than PIP2 interactions.
Corrected!
- Lines 204, 219. etc.: please explain what is meant by nonspecific nature of ML297 and GAT1587?
We have done so now in the revised manuscript: “Although ML297 was shown to be more biased towards GIRK1/2 activation than GIRK1/4 (~10-fold), it can be considered relatively nonselective when compared to GAT1508 (100-fold preference for GIRK1/2 over GIRK1/4 [19-20]”.
- Lines 230-232: “Notably, the methyl cyclopropyl ring interacts with GIRK4-W86 and not GIRK4-I190.” I190 is not labelled in the corresponding figure (Fig 2C, D). Should I195 be written there?
This was confusing the way it was presented. Only one of four interactions of the cyclopropyl ring of GAT1587 is retained in the GIRK1/4 case compared to the GIRK1/2 case and even that is not identical. The sentence has now been rephrased to clarify the point.
- Line 231: change confirmations to conformations.
Corrected!
- Section 2.4 title is long and complicated. Consider rephrasing.
The title of section 2.4 was shortened.
- Please check typos in lines 399-402; 410-416 (typos, commas missing, sentence about D173 is unclear); 473-484; line 526 (what do you mean by “These two transmembrane (TM) per subunit channels”?
The “two transmembrane per subunit” has been re-written for clarification. The sentence describing the two acidic residues that the hydrophobic wire couples to was broken into two sentences for clarity.
- Section 2.5 title is very long and convoluted.
We have rephrased the Section 2.5 title “Upon ML297 binding, TM1 movement is transduced to the Slide Helix (SH) and the CD loop, freeing GIRK1-K188 away from GIRK2-E315 (G-loop) and toward PIP2 binding”.
- Line 724: change from to form.
Corrected.

Reviewer 2 Report
Summary:
GIRK channels are hetero-tetrameric K+ channels that play functional roles in many organs including neurons and heart. Gazgalis and coauthors have studied the molecular mechanisms underlying the activation or inhibition of heteromeric GIRK1/2 and GIRK1/4 channels by two small molecules: ML297, which is an activator of GIRK1/2 and GIRK1/4 channels, and GAT1587, which acts as an inhibitor. Intriguingly, GAT1587 differs from ML297 only by the addition of a methyl cyclopropyl group in ML297, which switches the molecular effect from activation to inhibition.
The authors generated homology models of GIRK1/2 and GIRK1/4 channels and studied them by molecular dynamics simulations which allowed the observation of K+ permeation and protein dynamics related to channel gating. While ML297 was observed to increase K+ permeation, dilate the pore at the G-loop and helix bundle crossing gates, and increase the frequency of salt bridge interactions with PIP2, GAT1587 led to the opposite effects.
By comparing the MD trajectories of ML297-bound and GAT1587-bound channels the authors could identify distinct molecular interaction fingerprints which could be responsible for the distinct effects of ML297 and GAT1587. The results provide very detailed insights into gating mechanisms in GIRK channels.
This is a thoroughly performed and carefully analyzed simulation study. While it is not an absolute requirement, I think, the conclusions could be corroborated by a combination with experimental electrophysiology to test some predictions of the simulations.
Major comments:
- It is described that molecular models of GIRK1/2 and GIRK1/4 with ML297 and GAT1587 were generated by molecular docking. It was observed that ML297 binds between TM1 and TM2 of the GIRK1 subunit, while GAT1587 binds at the protein-lipid interface rather than intercalating between TM1 and TM2. There is some uncertainty in computational docking predictions which will affect subsequent MD results. Can the authors provide some validation how reliable their computationally predicted binding poses are?
- Did the authors find the same kinds of observations across all three MD replicates?
- For graphs showing recorded distances no unit of distance in given. I assume that authors show distances in Å and not nanometer or some other unit, but this is not 100% clear. This applies to all figures 1-5.
Minor comments:
I have a few methodological questions.
- Can the authors specify if peudo-C4-symmetry was considered in homology modeling?
- Maybe I missed it, but can the authors specify how many GIRK subunits were occupied by ML297 or GAG1587?
- Can the authors please specify which force field parameters they have used for the K+ ions to allow efficient K+ permeation? Did they have to apply any NB fix?
- Can the authors please provide a few more details in Methods on how they generated the constant voltage in their MD simulations with AMBER?
- How is it possible that there were several thousand contacts between TM1 and TM2 shown in the histograms in Figure 3E? How did the author define a contact?
I have noticed a few typos:
- Line 103 “from an inhibited to and activated conformation” should be “from an inhibited to an activated conformation”
- Line 243: “hydrophobic chain or residues in TM1” should be “hydrophobic chain of residues in TM1”
- Line 248: “With the Kir3 family” should be “Within the Kir3 family”
- Line 299: “Corelates” should be “Correlates”
- Line 350: “Van der Walls” should be “Van der Waals”
- Line 443: “ML297 induces and upward movement” should be “induces an upward movement”
- Line 674: “TIP3” should be “TIP3P”
- Line 702: “slate bridge” should be “salt bridge”
- Line 722: “Corelated network” should be “Correlated network”.
Author Response
REVIEWER 2
Summary:
GIRK channels are hetero-tetrameric K+ channels that play functional roles in many organs including neurons and heart. Gazgalis and coauthors have studied the molecular mechanisms underlying the activation or inhibition of heteromeric GIRK1/2 and GIRK1/4 channels by two small molecules: ML297, which is an activator of GIRK1/2 and GIRK1/4 channels, and GAT1587, which acts as an inhibitor. Intriguingly, GAT1587 differs from ML297 only by the addition of a methyl cyclopropyl group in ML297, which switches the molecular effect from activation to inhibition.
The authors generated homology models of GIRK1/2 and GIRK1/4 channels and studied them by molecular dynamics simulations which allowed the observation of K+ permeation and protein dynamics related to channel gating. While ML297 was observed to increase K+ permeation, dilate the pore at the G-loop and helix bundle crossing gates, and increase the frequency of salt bridge interactions with PIP2, GAT1587 led to the opposite effects.
By comparing the MD trajectories of ML297-bound and GAT1587-bound channels the authors could identify distinct molecular interaction fingerprints which could be responsible for the distinct effects of ML297 and GAT1587. The results provide very detailed insights into gating mechanisms in GIRK channels.
This is a thoroughly performed and carefully analyzed simulation study. While it is not an absolute requirement, I think, the conclusions could be corroborated by a combination with experimental electrophysiology to test some predictions of the simulations.
We thank the reviewer for the positive critique. We appreciate the suggestion to test experimentally predictions of the model and although we are already engaged in doing so in elegant ways (e.g., by employing photo-activated unnatural amino acids), we feel this is a long-term goal of our group and beyond the scope of the present manuscript. Please stay tuned…
Major comments:
- It is described that molecular models of GIRK1/2 and GIRK1/4 with ML297 and GAT1587 were generated by molecular docking. It was observed that ML297 binds between TM1 and TM2 of the GIRK1 subunit, while GAT1587 binds at the protein-lipid interface rather than intercalating between TM1 and TM2. There is some uncertainty in computational docking predictions which will affect subsequent MD results. Can the authors provide some validation how reliable their computationally predicted binding poses are?
Both ML297 and GAT1587 require the presence of GIRK1 to function. Due to the high degree of structural similarity of these compounds and the dependence on GIRK1, it is reasonable to expect that these compounds occupy a similar binding site. The evidence we have is mostly in the computational results presented in this study.
ML297 gates both GIRK1/2 and GIRK1/4 channels to allow K+ ion permeation and strengthen channel-PIP2 interactions. In contrast, GAT1587 inhibits both heteromeric channels by having the opposite effects on the collective variables we tested. These results are consistent with both prior computational and experimental studies. In 2019, we provided evidence for in silico GIRK2 gating by Gbg and Na+ ion in the presence of PIP2, a finding that reproduced experimental results published 20 years earlier (Li et al., 2019; Petit-Jacques et al., 1999; see references 5 and 14 in the revised manuscript). Even though the physiological and pharmacological channel modulators act at distinct sites, the mechanisms of allosteric control of channel-PIP2 interactions are shared. In 2020, we reported synthesis of a neuronal-specific variant of ML297, the compound GAT1508 that we characterized for its binding and gating effects both is silico and experimentally with electrophysiology (Xu et al., 2020; see reference 20 in the revised manuscript). The extensive mutagenesis provided by Xu and colleagues for GAT1508 validated further the binding site of urea-scaffold compounds and the mechanism by which they control the two channel gates by allosterically controlling channel-PIP2 interactions. Moreover, our present study explains how the two unique GIRK1 residues, F137 and D173, control gating by both ML297 and GAT1508.
We have added an opening paragraph to the discussion section of the revised manuscript to highlight these arguments that make us think that the computationally predicted binding sites are accurate. Atomic resolution structures of mammalian GIRK1-containing heteromers have been challenging to obtain as GIRK1 has been difficult to purify despite intense efforts for the past 20 years (7, 8). Thus, the present computational models provide alternative means of stimulating research towards mechanistic understanding and drug discovery efforts of GIRK1-containing heteromers [Lines 535-544].
- Did the authors find the same kinds of observations across all three MD replicates?
A number of replicas were used to confirm conformational changes. Results reported in this paper were seen in at least 2 of the 3 and at most 6 replicas. Examples of 6 replicas gated by ML297 and GAT1587 are shown in Figs. SF 4 and SF 5.
- For graphs showing recorded distances no unit of distance in given. I assume that authors show distances in Å and not nanometer or some other unit, but this is not 100% clear. This applies to all figures 1-5.
Units are in angstroms. The figures have been updated to reflect the units.
Minor comments:
I have a few methodological questions.
- Can the authors specify if peudo-C4-symmetry was considered in homology modeling?
Pseudo-C4-symmetry was not explicitly considered in the homology modeling. Instead, using Prime, we built the 4 subunits of the channel simultaneously.
- Maybe I missed it, but can the authors specify how many GIRK subunits were occupied by ML297 or GAT1587?
Since urea-based activators/inhibitors act on GIRK1 subunits (19,20) and GIRK1-heterotetramers have two GIRK1 subunits we used two molecules of ML297 or GAT1587 docked on each of the two GIRK1 subunits. This has now been clarified in the revised manuscript [lines 236-238].
- Can the authors please specify which force field parameters they have used for the K+ ions to allow efficient K+ permeation? Did they have to apply any NB fix?
The TIP3P leaprc file that is distributed with amber tools contains the force field parameters and non-bonded corrections for potassium ions and the solvent model. The methods section was updated to include the water model and ion parameters.
- Can the authors please provide a few more details in Methods on how they generated the constant voltage in their MD simulations with AMBER?
The constant voltage is generated through an applied electric field using the efz keyword on the input parameters. This keyword specifically defines an electric potential along the z-axis of the system. Given that we pre-aligned each model, the orientation of the 4KFM protein structure in the membrane, moving from positive z to negative z, represents going from the extracellular side of the plasma membrane, through the plasma membrane, and entering the intracellular side of the plasma membrane. A negative electrostatic potential would drive positively charged ions from positive z to negative z. Finally, an electric field of 0.06 kcal/(mol*A*e) corresponds to approximately 120mV across the plasma membrane which maximizes driving force near physiological membrane potential boundaries when the channel opens. [Lines 736-746]
- How is it possible that there were several thousand contacts between TM1 and TM2 shown in the histograms in Figure 3E? How did the author define a contact?
CPPTRAJ has a function that can be accessed with the native contacts keyword. Two groups of atoms can be defined and tracked using a simple distance cutoff. Native contacts are defined on a per-atom basis where one contact is 2 atoms below the cutoff distance. To make these contacts more meaningful, the number of contacts can be summed on a per residue per frame or group of residues per-frame basis. The default cutoff for this is 7 angstroms.
I have noticed a few typos:
- Line 103 “from an inhibited to and activated conformation” should be “from an inhibited to an activated conformation”
Corrected.
- Line 243: “hydrophobic chain or residues in TM1” should be “hydrophobic chain of residues in TM1”
Corrected.
- Line 248: “With the Kir3 family” should be “Within the Kir3 family”
Corrected.
- Line 299: “Corelates” should be “Correlates”
Corrected.
- Line 350: “Van der Walls” should be “Van der Waals”
Corrected.
- Line 443: “ML297 induces and upward movement” should be “induces an upward movement”
Corrected.
- Line 674: “TIP3” should be “TIP3P”
Corrected.
- Line 702: “slate bridge” should be “salt bridge”
Corrected.
- Line 722: “Corelated network” should be “Correlated network”.
Corrected.

Reviewer 3 Report
Typically, I tend to reject "pure" computational studies not backed up by any experiment - yet, this work is in my eyes different because the submicrosecond time scale may be just enough to make it relevant - because clear differences between apo and activated/inhibited channels seem to emerge. May - of course, it'll be better if authors could somehow experimentally validate their findings, but the invested computational effort is large enough to deserve publication. I don't see any technical problems - except, (and important) box plots showing means of calculated parameters should be augmented with stddev bars (and, perhaps, some statistical significance tests of the hypothesis that distributions of calculated magnitudes are significantly different - by Student test, or whatever - for apo, inhibited and activated proteins. This seems to be the case - but never trust your eyes). As far as "interest for the readers" I marked "average" because I'm not an ion channel person - therefore, this article should in my opinion also be reviewed by some wet lab experts in the field (structural biologists!) in order to see whether they find it inspiring.
Author Response
Reviewer 3
Typically, I tend to reject "pure" computational studies not backed up by any experiment - yet, this work is in my eyes different because the sub-microsecond time scale may be just enough to make it relevant - because clear differences between apo and activated/inhibited channels seem to emerge. May - of course, it'll be better if authors could somehow experimentally validate their findings, but the invested computational effort is large enough to deserve publication. I don't see any technical problems - except, (and important) box plots showing means of calculated parameters should be augmented with std dev bars (and, perhaps, some statistical significance tests of the hypothesis that distributions of calculated magnitudes are significantly different - by Student test, or whatever - for apo, inhibited and activated proteins. This seems to be the case - but never trust your eyes). As far as "interest for the readers" I marked "average" because I'm not an ion channel person - therefore, this article should in my opinion also be reviewed by some wet lab experts in the field (structural biologists!) in order to see whether they find it inspiring.
We thank the reviewer for the positive comments. This study also comes from a predominantly experimental lab. What compelled us to invest as much time and present a computational study alone are that 1) there is strong alignment of the computational results with experimental ones (e.g., the effects on all collective variables by the two types of ligands) and 2) that the computational experiment has a measurable end point (i.e., the ion channels are gated and ions permeate) serving as their own control.
Doing a statistical analysis suggested was nontrivial. To reliably statistically test these systems, we needed to conduct a power analysis to estimate the number of simulations needed. If we focus on the reporters with the largest statistical deviations, the GIRK1/2 activated G loop gate has the largest variability at approximately 2 angstroms. The mean of this population hovers between 7 - 8 angstroms. In the closed state, the population mean of this reporter is approximately 4 angstroms. Using these numbers to conduct a power analysis revealed that 6 samples per group would be required to conduct an independent sample T-test or ANOVA. It should be noted for this study, a one-way ANOVA was used to test for differences between the populations. However, since there are only 2 groups, the one-way ANOVA behaves like an independent sample T-test. The original purpose of the replicas used in this study was to test for convergence across multiple simulations. This was expanded to now use multiple poses for the initial coordinates rather than just varying the initial seed to generate random forces. The next 6 highest ranking poses were taken from our docking results and used to generate additional systems. These systems with lower ranking poses were then also run out to 300 ns using the same methodology as previously discussed in the methods section of the paper. What we are testing here is not just convergence on the reporters, but also validating the methodology used in this paper. We have reported the RMSD and channel gate distances on a per simulation basis in the supplemental material. These situations were also pooled together for further analysis. What is typically seen here is that ML297 includes both gates to open on the studies of the two systems. GAT1587 induces a constriction on these gates. Over the course of the simulation, one can see how the populations for each gate diverge in the presence of ML297 or GAT1587. Assuming that the gate distances are normally distributed, we can take the last 30 frames (~5ns) of each simulation for use in our statistical analysis. Conducting the statistical analysis, in all cases, we achieved P values of <0.0001. If we pool the normalized salt bridge formation and conduct similar statistical tests, we achieved P values of <0.001. Ion conduction was not included in this statistical analysis, as we focused on the changing conformational state. In addition, conduction was heavily tracked with the channel gate distances.
One further note to make regarding these expanded simulations is that due to the nature of dynamic simulations, the individual frames that make up a trajectory are correlated to one another. Using a single trajectory to conduct a statistical analysis is not a well-grounded idea due to this fact. Statistical tests, in most cases, require uncorrelated samples to be held valid. There are simulations such as Monte Carlo that give an uncorrelated sampling of the system, but here we focused on the specific changes that lead to channel activation and inhibition. Devising statistics from a dynamic simulation requires either that very long-time steps are placed between the frames of the trajectory or multiple replicas are run. In this study, we needed to run replicas. However, when you have a large system such as a heterotrimeric GIRK channel, this comes at a great computational expense. To gather the additional work that is presented in the revision, the additional replicas expanded the data included in this paper by a factor of 5 to 6. We wanted to be rigorous in our response and we hope the reviewer is satisfied with our attempt.
Figures SF 11 and SF 12 and Table ST 1 show the results of this statistical analysis.
